# Kernel Bayesian Inference with Posterior Regularization

**Yang Song**[†], **Jun Zhu**[‡][*], **Yong Ren**[‡]

[†] Dept. of Physics, Tsinghua University, Beijing, China
[‡] Dept. of Comp. Sci. & Tech., TNList Lab; Center for Bio-Inspired Computing Research
State Key Lab for Intell. Tech. & Systems, Tsinghua University, Beijing, China
`yangsong@cs.stanford.edu; {dcszj@, renyong15@mails}.tsinghua.edu.cn`

## Abstract

We propose a vector-valued regression problem whose solution is equivalent to the reproducing kernel Hilbert space (RKHS) embedding of the Bayesian posterior distribution. This equivalence provides a new understanding of kernel Bayesian inference. Moreover, the optimization problem induces a new regularization for the posterior embedding estimator, which is faster and has comparable performance to the squared regularization in kernel Bayes' rule. This regularization coincides with a former thresholding approach used in kernel POMDPs whose consistency remains to be established. Our theoretical work solves this open problem and provides consistency analysis in regression settings. Based on our optimizational formulation, we propose a flexible Bayesian posterior regularization framework which for the first time enables us to put regularization at the distribution level. We apply this method to nonparametric state-space filtering tasks with extremely nonlinear dynamics and show performance gains over all other baselines.

## 1 Introduction

Kernel methods have long been effective in generalizing linear statistical approaches to nonlinear cases by embedding a sample to the reproducing kernel Hilbert space (RKHS) [1]. In recent years, the idea has been generalized to embedding probability distributions [2, 3]. Such embeddings of probability measures are usually called *kernel embeddings* (a.k.a. *kernel means*). Moreover, [4, 5, 6] show that statistical operations of distributions can be realized in RKHS by manipulating kernel embeddings via linear operators. This approach has been applied to various statistical inference and learning problems, including training hidden Markov models (HMM) [7], belief propagation (BP) in tree graphical models [8], planning Markov decision processes (MDP) [9] and partially observed Markov decision processes (POMDP) [10].

One of the key workhorses in the above applications is the *kernel Bayes' rule* [5], which establishes the relation among the RKHS representations of the priors, likelihood functions and posterior distributions. Despite empirical success, the characterization of kernel Bayes' rule remains largely incomplete. For example, it is unclear how the estimators of the posterior distribution embeddings relate to optimizers of some loss functions, though the vanilla Bayes' rule has a nice connection [11]. This makes generalizing the results especially difficult and hinters the intuitive understanding of kernel Bayes' rule.

To alleviate this weakness, we propose a vector-valued regression [12] problem whose optimizer is the posterior distribution embedding. This new formulation is inspired by the progress in two fields: 1) the alternative characterization of conditional embeddings as regressors [13], and 2) the

---

[*]Corresponding author.

introduction of posterior regularized Bayesian inference (RegBayes) [14] based on an optimizational reformulation of the Bayes' rule.

We demonstrate the novelty of our formulation by providing a new understanding of kernel Bayesian inference, with theoretical, algorithmic and practical implications. On the theoretical side, we are able to prove the (weak) consistency of the estimator obtained by solving the vector-valued regression problem under reasonable assumptions. As a side product, our proof can be applied to a thresholding technique used in [10], whose consistency is left as an open problem. On the algorithmic side, we propose a new regularization technique, which is shown to run faster and has comparable accuracy to squared regularization used in the original kernel Bayes' rule [5]. Similar in spirit to RegBayes, we are also able to derive an extended version of the embeddings by directly imposing regularization on the posterior distributions. We call this new framework kRegBayes. Thanks to RKHS embeddings of distributions, this is the first time, to the best of our knowledge, people can do posterior regularization without invoking linear functionals (such as moments) of the random variables. On the practical side, we demonstrate the efficacy of our methods on both simple and complicated synthetic state-space filtering datasets.

Same to other algorithms based on kernel embeddings, our kernel regularized Bayesian inference framework is nonparametric and general. The algorithm is nonparametric, because the priors, posterior distributions and likelihood functions are all characterized by weighted sums of data samples. Hence it does not need the explicit mechanism such as differential equations of a robot arm in filtering tasks. It is general in terms of being applicable to a broad variety of domains as long as the kernels can be defined, such as strings, orthonormal matrices, permutations and graphs.

## 2 Preliminaries

### 2.1 Kernel embeddings

Let $(\mathcal{X}, \mathcal{B}_{\mathcal{X}})$ be a measurable space of random variables, $p_X$ be the associated probability measure and $\mathcal{H}_{\mathcal{X}}$ be a RKHS with kernel $k(\cdot, \cdot)$. We define the *kernel embedding* of $p_X$ to be $\mu_X = \mathbb{E}_{p_X}[\phi(X)] \in \mathcal{H}_{\mathcal{X}}$, where $\phi(X) = k(X, \cdot)$ is the feature map. Such a vector-valued expectation always exists if the kernel is bounded, namely $\sup_x k_{\mathcal{X}}(x, x) < \infty$.

The concept of kernel embeddings has several important statistical merits. Inasmuch as the reproducing property, the expectation of $f \in \mathcal{H}$ w.r.t. $p_X$ can be easily computed as $\mathbb{E}_{p_X}[f(X)] = \mathbb{E}_{p_X}[\langle f, \phi(X) \rangle] = \langle f, \mu_X \rangle$. There exists *universal kernels* [15] whose corresponding RKHS $\mathcal{H}$ is dense in $\mathcal{C}_{\mathcal{X}}$ in terms of sup norm. This means $\mathcal{H}$ contains a rich range of functions $f$ and their expectations can be computed by inner products without invoking usually intractable integrals. In addition, the inner product structure of the embedding space $\mathcal{H}$ provides a natural way to measure the differences of distributions through norms.

In much the same way we can define kernel embeddings of linear operators. Let $(\mathcal{X}, \mathcal{B}_{\mathcal{X}})$ and $(\mathcal{Y}, \mathcal{B}_{\mathcal{Y}})$ be two measurable spaces, $\phi(x)$ and $\psi(y)$ be the measurable feature maps of corresponding RKHS $\mathcal{H}_{\mathcal{X}}$ and $\mathcal{H}_{\mathcal{Y}}$ with bounded kernels, and $p$ denote the joint distribution of a random variable $(X, Y)$ on $\mathcal{X} \times \mathcal{Y}$ with product measures. The *covariance operator* $\mathcal{C}_{XY}$ is defined as $\mathcal{C}_{XY} = \mathbb{E}_p[\phi(X) \otimes \psi(Y)]$, where $\otimes$ denotes the tensor product. Note that it is possible to identify $\mathcal{C}_{XY}$ with $\mu_{(XY)}$ in $\mathcal{H}_{\mathcal{X}} \otimes \mathcal{H}_{\mathcal{Y}}$ with the kernel function $k((x_1, y_1), (x_2, y_2)) = k_{\mathcal{X}}(x_1, x_2) k_{\mathcal{Y}}(y_1, y_2)$ [16]. There is an important relation between kernel embeddings of distributions and covariance operators, which is fundamental for the sequel:

**Theorem 1** ([4, 5]). *Let $\mu_X$, $\mu_Y$ be the kernel embeddings of $p_X$ and $p_Y$ respectively. If $\mathcal{C}_{XX}$ is injective, $\mu_X \in \mathcal{R}(\mathcal{C}_{XX})$ and $\mathbb{E}[g(Y) \mid X = \cdot] \in \mathcal{H}_{\mathcal{X}}$ for all $g \in \mathcal{H}_{\mathcal{Y}}$, then*

$$\mu_Y = \mathcal{C}_{YX} \mathcal{C}_{XX}^{-1} \mu_X. \tag{1}$$

*In addition, $\mu_{Y|X=x} = \mathbb{E}[\psi(Y)|X = x] = \mathcal{C}_{YX} \mathcal{C}_{XX}^{-1} \phi(x)$.*

On the implementation side, we need to estimate these kernel embeddings via samples. An intuitive estimator for the embedding $\mu_X$ is $\widehat{\mu}_X = \frac{1}{N} \sum_{i=1}^{N} \phi(x_i)$, where $\{x_i\}_{i=1}^{N}$ is a sample from $p_X$. Similarly, the covariance operators can also be estimated by $\widehat{\mathcal{C}}_{XY} = \frac{1}{N} \sum_{i=1}^{N} \phi(x_i) \otimes \psi(y_i)$. Both operators are shown to converge in the RKHS norm at a rate of $O_p(N^{-\frac{1}{2}})$ [4].

## 2.2 Kernel Bayes' rule

Let $\pi(Y)$ be the prior distribution of a random variable $Y$, $p(X = x \mid Y)$ be the likelihood, $p^\pi(Y \mid X = x)$ be the posterior distribution given $\pi(Y)$ and observation $x$, and $p^\pi(X, Y)$ be the joint distribution incorporating $\pi(Y)$ and $p(X \mid Y)$. Kernel Bayesian inference aims to obtain the posterior embedding $\mu_Y^\pi(X = x)$ given a prior embedding $\pi_Y$ and a covariance operator $\mathcal{C}_{XY}$. By Bayes' rule, $p^\pi(Y \mid X = x) \propto \pi(Y)p(X = x \mid Y)$. We assume that there exists a joint distribution $p$ on $\mathcal{X} \times \mathcal{Y}$ whose conditional distribution matches $p(X \mid Y)$ and let $\mathcal{C}_{XY}$ be its covariance operator. Note that we do not require $p = p^\pi$ hence $p$ can be any convenient distribution.

According to Thm. 1, $\mu_Y^\pi(X = x) = \mathcal{C}_{YX}^\pi \mathcal{C}_{XX}^{\pi}{}^{-1}\phi(x)$, where $\mathcal{C}_{YX}^\pi$ corresponds to the joint distribution $p^\pi$ and $\mathcal{C}_{XX}^\pi$ to the marginal probability of $p^\pi$ on $X$. Recall that $\mathcal{C}_{YX}^\pi$ can be identified with $\mu_{(YX)}$ in $\mathcal{H}_\mathcal{Y} \otimes \mathcal{H}_\mathcal{X}$, we can apply Thm. 1 to obtain $\mu_{(YX)} = \mathcal{C}_{(YX)Y}\mathcal{C}_{YY}^{-1}\pi_Y$, where $\mathcal{C}_{(YX)Y} := \mathbb{E}[\psi(Y) \otimes \phi(X) \otimes \psi(Y)]$. Similarly, $\mathcal{C}_{XX}^\pi$ can be represented as $\mu_{(XX)} = \mathcal{C}_{(XX)Y}\mathcal{C}_{YY}^{-1}\pi_Y$. This way of computing posterior embeddings is called the *kernel Bayes' rule* [5].

Given estimators of the prior embedding $\widehat{\pi}_Y = \sum_{i=1}^m \tilde{\alpha}_i \psi(y_i)$ and the covariance operator $\widehat{\mathcal{C}}_{YX}$, The posterior embedding can be obtained via $\widehat{\mu}_Y^\pi(X = x) = \widehat{\mathcal{C}}_{YX}^\pi([\widehat{\mathcal{C}}_{XX}^\pi]^2 + \lambda I)^{-1}\widehat{\mathcal{C}}_{XX}^\pi \phi(x)$, where squared regularization is added to the inversion. Note that the regularization for $\widehat{\mu}_Y^\pi(X = x)$ is not unique. A thresholding alternative is proposed in [10] without establishing the consistency. We will discuss this thresholding regularization in a different perspective and give consistency results in the sequel.

## 2.3 Regularized Bayesian inference

Regularized Bayesian inference (RegBayes [14]) is based on a variational formulation of the Bayes' rule [11]. The posterior distribution can be viewed as the solution of $\min_{p(Y|X=x)} \mathrm{KL}(p(Y|X = x)\|\pi(Y)) - \int \log p(X = x|Y)\mathrm{d}p(Y|X = x)$, subjected to $p(Y|X = x) \in \mathcal{P}_{\mathrm{prob}}$, where $\mathcal{P}_{\mathrm{prob}}$ is the set of valid probability measures. RegBayes combines this formulation and posterior regularization [17] in the following way

$$\min_{p(Y|X=x),\xi} \mathrm{KL}(p(Y|X = x)\|\pi(Y)) - \int \log p(X = x|Y)\mathrm{d}p(Y|X = x) + U(\xi)$$
$$s.t. \quad p(Y|X = x) \in \mathcal{P}_{\mathrm{prob}}(\xi),$$

where $\mathcal{P}_{\mathrm{prob}}(\xi)$ is a subset depending on $\xi$ and $U(\xi)$ is a loss function. Such a formulation makes it possible to regularize Bayesian posterior distributions, smoothing the gap between Bayesian generative models and discriminative models. Related applications include max-margin topic models [18] and infinite latent SVMs [14].

Despite the flexibility of RegBayes, regularization on the posterior distributions is practically imposed indirectly via expectations of a function. We shall see soon in the sequel that our new framework of kernel Regularized Bayesian inference can control the posterior distribution in a direct way.

## 2.4 Vector-valued regression

The main task for vector-valued regression [12] is to minimize the following objective

$$E(f) := \sum_{i=1}^n \|y_j - f(x_j)\|_{\mathcal{H}_\mathcal{Y}}^2 + \lambda \|f\|_{\mathcal{H}_K}^2,$$

where $y_j \in \mathcal{H}_\mathcal{Y}$, $f : \mathcal{X} \to \mathcal{H}_\mathcal{Y}$. Note that $f$ is a function with RKHS values and we assume that $f$ belongs to a *vector-valued* RKHS $\mathcal{H}_K$. In vector-valued RKHS, the kernel function $k$ is generalized to linear operators $\mathcal{L}(\mathcal{H}_\mathcal{Y}) \ni K(x_1, x_2) : \mathcal{H}_\mathcal{Y} \to \mathcal{H}_\mathcal{Y}$, such that $K(x_1, x_2)y := (K_{x_2}y)(x_1)$ for every $x_1, x_2 \in \mathcal{X}$ and $y \in \mathcal{H}_\mathcal{Y}$, where $K_{x_2}y \in \mathcal{H}_K$. The reproducing property is generalized to $\langle y, f(x)\rangle_{\mathcal{H}_\mathcal{Y}} = \langle K_x y, f\rangle_{\mathcal{H}_K}$ for every $y \in \mathcal{H}_\mathcal{Y}$, $f \in \mathcal{H}_K$ and $x \in \mathcal{X}$. In addition, [12] shows that the representer theorem still holds for vector-valued RKHS.

## 3 Kernel Bayesian inference as a regression problem

One of the unique merits of the posterior embedding $\mu_Y^\pi(X = x)$ is that expectations w.r.t. posterior distributions can be computed via inner products, *i.e.*, $\langle h, \mu_Y^\pi(X = x)\rangle = \mathbb{E}_{p^\pi(Y|X=x)}[h(Y)]$ for all

$h \in \mathcal{H}_\mathcal{Y}$. Since $\mu_Y^\pi(X = x) \in \mathcal{H}_\mathcal{Y}$, $\mu_Y^\pi$ can be viewed as an element of a vector-valued RKHS $\mathcal{H}_K$ containing functions $f : \mathcal{X} \to \mathcal{H}_\mathcal{Y}$.

A natural optimization objective [13] thus follows from the above observations

$$\mathcal{E}[\mu] := \sup_{\|h\|_\mathcal{Y} \leq 1} \mathbb{E}_X \left[ (\mathbb{E}_Y[h(Y)|X] - \langle h, \mu(X) \rangle_{\mathcal{H}_\mathcal{Y}})^2 \right], \tag{2}$$

where $\mathbb{E}_X[\cdot]$ denotes the expectation w.r.t. $p^\pi(X)$ and $\mathbb{E}_Y[\cdot|X]$ denotes the expectation w.r.t. the Bayesian posterior distribution, *i.e.*, $p^\pi(Y \mid X) \propto \pi(Y)p(X \mid Y)$. Clearly, $\mu_Y^\pi = \arg\inf_\mu \mathcal{E}[\mu]$. Following [13], we introduce an upper bound $\mathcal{E}_s$ for $\mathcal{E}$ by applying Jensen's and Cauchy-Schwarz's inequalities consecutively

$$\mathcal{E}_s[\mu] := \mathbb{E}_{(X,Y)}[\|\psi(Y) - \mu(X)\|_{\mathcal{H}_\mathcal{Y}}^2], \tag{3}$$

where $(X, Y)$ is the random variable on $\mathcal{X} \times \mathcal{Y}$ with the joint distribution $p^\pi(X, Y) = \pi(Y)p(X \mid Y)$.

The first step to make this optimizational framework practical is to find finite sample estimators of $\mathcal{E}_s[\mu]$. We will show how to do this in the following section.

## 3.1 A consistent estimator of $\mathcal{E}_s[\mu]$

Unlike the conditional embeddings in [13], we do not have i.i.d. samples from the joint distribution $p^\pi(X, Y)$, as the priors and likelihood functions are represented with samples from different distributions. We will eliminate this problem using a kernel trick, which is one of our main innovations in this paper.

The idea is to use the inner product property of a kernel embedding $\mu_{(X,Y)}$ to represent the expectation $\mathbb{E}_{(X,Y)}[\|\psi(Y) - \mu(X)\|_{\mathcal{H}_\mathcal{Y}}^2]$ and then use finite sample estimators of $\mu_{(X,Y)}$ to estimate $\mathcal{E}_s[\mu]$. Recall that we can identify $\mathcal{C}_{XY} := \mathbb{E}_{XY}[\phi(X) \otimes \psi(Y)]$ with $\mu_{(X,Y)}$ in a product space $\mathcal{H}_\mathcal{X} \otimes \mathcal{H}_\mathcal{Y}$ with a product kernel $k_\mathcal{X} k_\mathcal{Y}$ on $\mathcal{X} \times \mathcal{Y}$ [16]. Let $f(x, y) = \|\psi(y) - \mu(x)\|_{\mathcal{H}_\mathcal{Y}}^2$ and assume that $f \in \mathcal{H}_\mathcal{X} \otimes \mathcal{H}_\mathcal{Y}$. The optimization objective $\mathcal{E}_s[\mu]$ can be written as

$$\mathcal{E}_s[\mu] = \mathbb{E}_{(X,Y)}[\|\psi(Y) - \mu(X)\|_{\mathcal{H}_\mathcal{Y}}^2] = \langle f, \mu_{(X,Y)} \rangle_{\mathcal{H}_\mathcal{X} \otimes \mathcal{H}_\mathcal{Y}}. \tag{4}$$

From Thm. 1, we assert that $\mu_{(X,Y)} = \mathcal{C}_{(X,Y)Y} \mathcal{C}_{YY}^{-1} \pi_Y$ and a natural estimator follows to be $\widehat{\mu}_{(X,Y)} = \widehat{\mathcal{C}}_{(X,Y)Y}(\widehat{\mathcal{C}}_{YY} + \lambda I)^{-1}\widehat{\pi}_Y$. As a result, $\widehat{\mathcal{E}}_s[\mu] := \langle \widehat{\mu}_{(X,Y)}, f \rangle_{\mathcal{H}_\mathcal{X} \otimes \mathcal{H}_\mathcal{Y}}$ and we introduce the following proposition to write $\widehat{\mathcal{E}}_s$ in terms of Gram matrices.

**Proposition 1** (Proof in Appendix). *Suppose $(X, Y)$ is a random variable in $\mathcal{X} \times \mathcal{Y}$, where the prior for $Y$ is $\pi(Y)$ and the likelihood is $p(X \mid Y)$. Let $\mathcal{H}_\mathcal{X}$ be a RKHS with kernel $k_\mathcal{X}$ and feature map $\phi(x)$, $\mathcal{H}_\mathcal{Y}$ be a RKHS with kernel $k_\mathcal{Y}$ and feature map $\psi(y)$, $\phi(x, y)$ be the feature map of $\mathcal{H}_\mathcal{X} \otimes \mathcal{H}_\mathcal{Y}$, $\widehat{\pi}_Y = \sum_{i=1}^l \tilde{\alpha}_i \psi(\tilde{y}_i)$ be a consistent estimator of $\pi_Y$ and $\{(x_i, y_i)\}_{i=1}^n$ be a sample representing $p(X \mid Y)$. Under the assumption that $f(x, y) = \|\psi(y) - \mu(x)\|_{\mathcal{H}_\mathcal{Y}}^2 \in \mathcal{H}_\mathcal{X} \otimes \mathcal{H}_\mathcal{Y}$, we have*

$$\widehat{\mathcal{E}}_s[\mu] = \sum_{i=1}^n \beta_i \|\psi(y_i) - \mu(x_i)\|_{\mathcal{H}_\mathcal{Y}}^2, \tag{5}$$

*where $\boldsymbol{\beta} = (\beta_1, \cdots, \beta_n)^\intercal$ is given by $\boldsymbol{\beta} = (G_Y + n\lambda I)^{-1}\tilde{G}_Y \tilde{\boldsymbol{\alpha}}$, where $(G_Y)_{ij} = k_\mathcal{Y}(y_i, y_j)$, $(\tilde{G}_Y)_{ij} = k_\mathcal{Y}(y_i, \tilde{y}_j)$, and $\tilde{\boldsymbol{\alpha}} = (\tilde{\alpha}_1, \cdots, \tilde{\alpha}_l)^\intercal$.*

The consistency of $\widehat{\mathcal{E}}_s[\mu]$ is a direct consequence of the following theorem adapted from [5], since the Cauchy-Schwarz inequality ensures $|\langle \mu_{(X,Y)}, f \rangle - \langle \widehat{\mu}_{(X,Y)}, f \rangle| \leq \|\mu_{(X,Y)} - \widehat{\mu}_{(X,Y)}\| \|f\|$.

**Theorem 2** (Adapted from [5], Theorem 8). *Assume that $\mathcal{C}_{YY}$ is injective, $\widehat{\pi}_Y$ is a consistent estimator of $\pi_Y$ in $\mathcal{H}_\mathcal{Y}$ norm, and that $\mathbb{E}[k((X, Y), (\tilde{X}, \tilde{Y})) \mid Y = y, \tilde{Y} = \tilde{y}]$ is included in $\mathcal{H}_\mathcal{Y} \otimes \mathcal{H}_\mathcal{Y}$ as a function of $(y, \tilde{y})$, where $(\tilde{X}, \tilde{Y})$ is an independent copy of $(X, Y)$. Then, if the regularization coefficient $\lambda_n$ decays to $0$ sufficiently slowly,*

$$\left\| \widehat{\mathcal{C}}_{(X,Y)Y}(\widehat{\mathcal{C}}_{YY} + \lambda_n I)^{-1}\widehat{\pi}_Y - \mu_{(X,Y)} \right\|_{\mathcal{H}_\mathcal{X} \otimes \mathcal{H}_\mathcal{Y}} \to 0 \tag{6}$$

*in probability as $n \to \infty$.*

Although $\widehat{\mathcal{E}}_s[\mu]$ is a consistent estimator of $\mathcal{E}_s[\mu]$, it does not necessarily have minima, since the coefficients $\beta_i$ can be negative. One of our main contributions in this paper is the discovery that we can ignore data points $(x_i, y_i)$ with a negative $\beta_i$, *i.e.*, replacing $\beta_i$ with $\beta_i^+ := \max(0, \beta_i)$ in $\widehat{\mathcal{E}}_s[\mu]$. We will give explanations and theoretical justifications in the next section.

## 3.2    The thresholding regularization

We show in the following theorem that $\widehat{\mathcal{E}}_s^+[\mu] := \sum_{i=1}^n \beta_i^+ \|\psi(y_i) - \mu(x_i)\|^2$ converges to $\mathcal{E}_s[\mu]$ in probability in discrete situations. The trick of replacing $\beta_i$ with $\beta_i^+$ is named *thresholding regularization*.

**Theorem 3** (Proof in Appendix). *Assume that $\mathcal{X}$ is compact and $|\mathcal{Y}| < \infty$, $k$ is a strictly positive definite continuous kernel with $\sup_{(x,y)} k((x,y),(x,y)) < \kappa$ and $f(x,y) = \|\psi(y) - \mu(x)\|_{\mathcal{H}_\mathcal{Y}}^2 \in \mathcal{H}_\mathcal{X} \otimes \mathcal{H}_\mathcal{Y}$. With the conditions in Thm. 2, we assert that $\widehat{\mu}_{(X,Y)}^+$ is a consistent estimator of $\mu_{(X,Y)}$ and $\left|\widehat{\mathcal{E}}_s^+[\mu] - \mathcal{E}_s[\mu]\right| \to 0$ in probability as $n \to \infty$.*

In the context of partially observed Markov decision processes (POMDPs) [10], a similar thresholding approach, combined with normalization, was proposed to make the Bellman operator isotonic and contractive. However, the authors left the consistency of that approach as an open problem. The justification of normalization has been provided in [13], Lemma 2.2 under the finite space assumption. A slight modification of our proof of Thm. 3 (change the probability space from $\mathcal{X} \times \mathcal{Y}$ to $\mathcal{X}$) can complete the other half as a side product, under the same assumptions.

Compared to the original squared regularization used in [5], thresholding regularization is more computational efficient because 1) it does not need to multiply the Gram matrix twice, and 2) it does not need to take into consideration those data points with negative $\beta_i$'s. In many cases a large portion of $\{\beta_i\}_{i=1}^n$ is negative but the sum of their absolute values is small. The finite space assumption in Thm. 3 may also be weakened, but it requires deeper theoretical analyses.

## 3.3    Minimizing $\widehat{\mathcal{E}}_s^+[\mu]$

Following the standard steps of solving a RKHS regression problem, we add a Tikhonov regularization term to $\widehat{\mathcal{E}}_s^+[\mu]$ to provide a well-proposed problem,

$$\widehat{\mathcal{E}}_{\lambda,n}[\mu] = \sum_{i=1}^n \beta_i^+ \|\psi(y_i) - \mu(x_i)\|_{\mathcal{H}_\mathcal{Y}}^2 + \lambda \|\mu\|_{\mathcal{H}_K}^2 . \tag{7}$$

Let $\widehat{\mu}_{\lambda,n} = \arg\min_\mu \widehat{\mathcal{E}}_{\lambda,n}[\mu]$. Note that $\widehat{\mathcal{E}}_{\lambda,n}[\mu]$ is a vector-valued regression problem, and the representer theorems in vector-valued RKHS apply here. We summarize the matrix expression of $\widehat{\mu}_{\lambda,n}$ in the following proposition.

**Proposition 2** (Proof in Appendix). *Without loss of generality, we assume that $\beta_i^+ \neq 0$ for all $1 \leq i \leq n$. Let $\mu \in \mathcal{H}_K$ and choose the kernel of $\mathcal{H}_K$ to be $K(x_i, x_j) = k_\mathcal{X}(x_i, x_j)\mathcal{I}$, where $\mathcal{I} : \mathcal{H}_K \to \mathcal{H}_K$ is an identity map. Then*

$$\widehat{\mu}_{\lambda,n}(x) = \Psi(K_X + \lambda_n \Lambda^+)^{-1} K_{:x}, \tag{8}$$

*where $\Psi = (\psi(y_1), \cdots, \psi(y_n))$, $(K_X)_{ij} = k_\mathcal{X}(x_i, x_j)$, $\Lambda^+ = \mathrm{diag}(1/\beta_1^+, \cdots, 1/\beta_n^+)$, $K_{:x} = (k_\mathcal{X}(x, x_1), \cdots, k_\mathcal{X}(x, x_n))^\mathsf{T}$ and $\lambda_n$ is a positive regularization constant.*

## 3.4    Theoretical justifications for $\widehat{\mu}_{\lambda,n}$

In this section, we provide theoretical explanations for using $\widehat{\mu}_{\lambda,n}$ as an estimator of the posterior embedding under specific assumptions. Let $\mu^* = \arg\min_\mu \mathcal{E}[\mu]$, $\mu' = \arg\min_\mu \mathcal{E}_s[\mu]$, and recall that $\widehat{\mu}_{\lambda,n} = \arg\min_\mu \widehat{\mathcal{E}}_{\lambda,n}[\mu]$. We first show the relations between $\mu^*$ and $\mu'$ and then discuss the relations between $\widehat{\mu}_{\lambda,n}$ and $\mu'$.

The forms of $\mathcal{E}$ and $\mathcal{E}_s$ are exactly the same for posterior kernel embeddings and conditional kernel embeddings. As a consequence, the following theorem in [13] still hold.

**Theorem 4** ([13]). *If there exists a $\mu^* \in \mathcal{H}_K$ such that for any $h \in \mathcal{H}_{\mathcal{Y}}$, $\mathbb{E}[h|X] = \langle h, \mu^*(X) \rangle_{\mathcal{H}_{\mathcal{Y}}}$ $p_X$-a.s., then $\mu^*$ is the $p_X$-a.s. unique minimiser of both objectives:*

$$\mu^* = \underset{\mu \in \mathcal{H}_K}{\arg\min} \, \mathcal{E}[\mu] = \underset{\mu \in \mathcal{H}_K}{\arg\min} \, \mathcal{E}_s[\mu].$$

This theorem shows that if the vector-valued RKHS $\mathcal{H}_K$ is rich enough to contain $\mu^{\pi}_{Y|X=x}$, both $\mathcal{E}$ and $\mathcal{E}_s$ can lead us to the correct embedding. In this case, it is reasonable to use $\mu'$ instead of $\mu^*$. For the situation where $\mu^{\pi}_{Y|X=x} \notin \mathcal{H}_K$, we refer the readers to [13].

Unfortunately, we cannot obtain the relation between $\widehat{\mu}_{\lambda,n}$ and $\mu'$ by referring to [19], as in [13]. The main difficulty here is that $\{(x_i, y_i)\}|_{i=1}^n$ is not an i.i.d. sample from $p^{\pi}(X, Y) = \pi(Y)p(X \mid Y)$ and the estimator $\widehat{\mathcal{E}}_s^+[\mu]$ does not use i.i.d. samples to estimate expectations. Therefore the concentration inequality ([19], Prop. 2) used in the proofs of [19] cannot be applied.

To solve the problem, we propose Thm. 9 (in Appendix) which can lead to a consistency proof for $\widehat{\mu}_{\lambda,n}$. The relation between $\widehat{\mu}_{\lambda,n}$ and $\mu'$ can now be summarized in the following theorem.

**Theorem 5** (Proof in Appendix). *Assume Hypothesis 1 and Hypothesis 2 in [20] and our Assumption 1 (in the Appendix) hold. With the conditions in Thm. 3, we assert that if $\lambda_n$ decreases to 0 sufficiently slowly,*

$$\mathcal{E}_s[\widehat{\mu}_{\lambda_n, n}] - \mathcal{E}_s[\mu'] \to 0 \tag{9}$$

*in probability as $n \to \infty$.*

## 4 Kernel Bayesian inference with posterior regularization

Based on our optimizational formulation of kernel Bayesian inference, we can add additional regularization terms to control the posterior embeddings. This technique gives us the possibility to incorporate rich side information from domain knowledge and to enforce supervisions on Bayesian inference. We call our framework of imposing posterior regularization *kRegBayes*.

As an example of the framework, we study the following optimization problem

$$\mathcal{L} := \underbrace{\sum_{i=1}^m \beta_i^+ \left\| \mu(x_i) - \psi(y_i) \right\|_{\mathcal{H}_{\mathcal{Y}}}^2 + \lambda \left\| \mu \right\|_{\mathcal{H}_K}^2}_{\widehat{\mathcal{E}}_{\lambda,n}[\mu]} + \delta \underbrace{\sum_{i=m+1}^n \left\| \mu(x_i) - \psi(t_i) \right\|_{\mathcal{H}_{\mathcal{Y}}}^2}_{\text{The regularization term}}, \tag{10}$$

where $\{(x_i, y_i)\}_{i=1}^m$ is the sample used for representing the likelihood, $\{(x_i, t_i)\}_{i=m+1}^n$ is the sample used for posterior regularization and $\lambda, \delta$ are the regularization constants. Note that in RKHS embeddings, $\psi(t)$ is identified as a point distribution at $t$ [2]. Hence the regularization term in (10) encourages the posterior distributions $p(Y \mid X = x_i)$ to be concentrated at $t_i$. More complicated regularization terms are also possible, such as $\|\mu(x_i) - \sum_{i=1}^l \alpha_i \psi(t_i)\|_{\mathcal{H}_{\mathcal{Y}}}$.

Compared to vanilla RegBayes, our kernel counterpart has several obvious advantages. First, the difference between two distributions can be naturally measured by RKHS norms. This makes it possible to regularize the posterior distribution as a whole, rather than through expectations of discriminant functions. Second, the framework of kernel Bayesian inference is totally nonparametric, where the priors and likelihood functions are all represented by respective samples. We will further demonstrate the properties of kRegBayes through experiments in the next section.

Let $\widehat{\mu}_{reg} = \arg\min_{\mu} \mathcal{L}$. It is clear that solving $\mathcal{L}$ is substantially the same as $\widehat{\mathcal{E}}_{\lambda,n}[\mu]$ and we summarize it in the following proposition.

**Proposition 3.** *With the conditions in Prop. 2, we have*

$$\widehat{\mu}_{reg}(x) = \Psi(K_X + \lambda \Lambda^+)^{-1} K_{:x}, \tag{11}$$

*where $\Psi = (\psi(y_1), \cdots, \psi(y_n))$, $(K_X)_{ij} = k_{\mathcal{X}}(x_i, x_j)|_{1 \le i,j \le n}$, $\Lambda^+ = \text{diag}(1/\beta_1^+, \cdots, 1/\beta_m^+, 1/\delta, \cdots, 1/\delta)$, and $K_{:x} = (k_{\mathcal{X}}(x, x_1), \cdots, k_{\mathcal{X}}(x, x_n))^{\intercal}$.*

# 5 Experiments

In this section, we compare the results of kRegBayes and several other baselines for two state-space filtering tasks. The mechanism behind kernel filtering is stated in [5] and we provide a detailed introduction in Appendix, including all the formula used in implementation.

**Toy dynamics** This experiment is a twist of that used in [5]. We report the results of extended Kalman filter (EKF) [21] and unscented Kalman filter (UKF) [22], kernel Bayes' rule (KBR) [5], kernel Bayesian learning with thresholding regularization (pKBR) and kRegBayes.

The data points $\{(\theta_t, x_t, y_t)\}$ are generated from the dynamics

$$\theta_{t+1} = \theta_t + 0.4 + \xi_t \pmod{2\pi}, \quad \begin{pmatrix} x_{t+1} \\ y_{t+1} \end{pmatrix} = (1 + \sin(8\theta_{t+1})) \begin{pmatrix} \cos\theta_{t+1} \\ \sin\theta_{t+1} \end{pmatrix} + \zeta_t, \qquad (12)$$

where $\theta_t$ is the hidden state, $(x_t, y_t)$ is the observation, $\xi_t \sim \mathcal{N}(0, 0.04)$ and $\zeta_t \sim \mathcal{N}(0, 0.04)$. Note that this dynamics is nonlinear for both transition and observation functions. The observation model is an oscillation around the unit circle. There are 1000 training data and 200 validation/test data for each algorithm.

We suppose that EKF, UKF and kRegBayes know the true dynamics of the model and the first hidden state $\theta_1$. In this case, we use $\tilde{\theta}_{t+1} = \theta_1 + 0.4t \pmod{2\pi}$ and $(\tilde{x}_{t+1}, \tilde{y}_{t+1})^{\mathsf{T}} = (1 + \sin(8\tilde{\theta}_{t+1}))(\cos\tilde{\theta}_{t+1}, \sin\tilde{\theta}_{t+1})^{\mathsf{T}}$ as the supervision data point for the $(t+1)$-th step. We follow [5] to set our parameters.

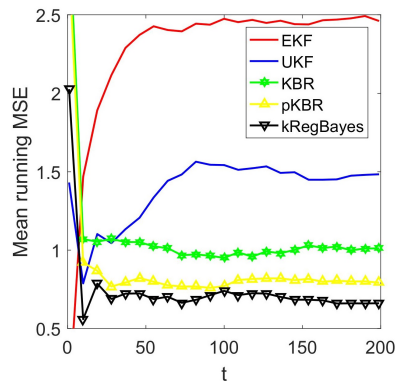

The results are summarized in Fig. 5. pKBR has lower errors compared to KBR, which means the thresholding regularization is practically no worse than the original squared regularization. The lower MSE of kRegBayes compared with pKBR shows that the posterior regularization successfully incorporates information from equations of the dynamics. Moreover, pKBR and kRegBayes run faster than KBR. The total running times for 50 random datasets of pKBR, kRegBayes and KBR are respectively 601.3s, 677.5s and 3667.4s.

Figure 1: Mean running MSEs against time steps for each algorithm. (Best view in color)

**Camera position recovery** In this experiment, we build a scene containing a table and a chair, which is derived from `classchair.pov` (http://www.oyonale.com). With a fixed focal point, the position of the camera uniquely determines the view of the scene. The task of this experiment is to estimate the position of the camera given the image. This is a problem with practical applications in remote sensing and robotics.

We vary the position of the camera in a plane with a fixed height. The transition equations of the hidden states are

$$\theta_{t+1} = \theta_t + 0.2 + \xi_\theta, \quad r_{t+1} = \max(R_2, \min(R_1, r_t + \xi_r)), \quad x_{t+1} = \cos\theta_{t+1}, \quad y_{t+1} = \sin\theta_{t+1},$$

where $\xi_\theta \sim \mathcal{N}(0, 4e-4)$, $\xi_r \sim \mathcal{N}(0, 1)$, $0 \leq R_1 < R_2$ are two constants and $\{(x_t, y_t)\}|_{t=1}^m$ are treated as the hidden variables. As the observation at $t$-th step, we render a $100 \times 100$ image with the camera located at $(x_t, y_t)$. For training data, we set $R_1 = 0$ and $R_2 = 10$ while for validation data and test data we set $R_1 = 5$ and $R_2 = 7$. The motivation is to distinguish the efficacy of enforcing the posterior distribution to concentrate around distance 6 by kRegBayes. We show a sample set of training and test images in Fig. 2.

We compare KBR, pKBR and kRegBayes with the traditional linear Kalman filter (KF [23]). Following [4] we down-sample the images and train a linear regressor for observation model. In all experiments, we flatten the images to a column vector and apply Gaussian RBF kernels if needed. The kernel band widths are set to be the median distances in the training data. Based on experiments on the validation dataset, we set $\lambda_T = 1e-6 = 2\delta_T$ and $\mu_T = 1e-5$.

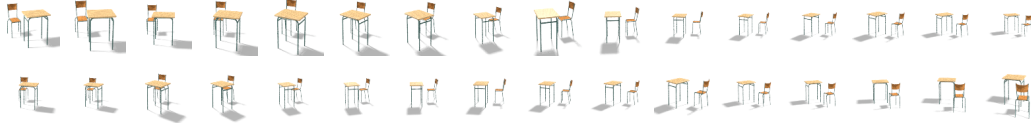

Figure 2: First several frames of training data (upper row) and test data (lower row).

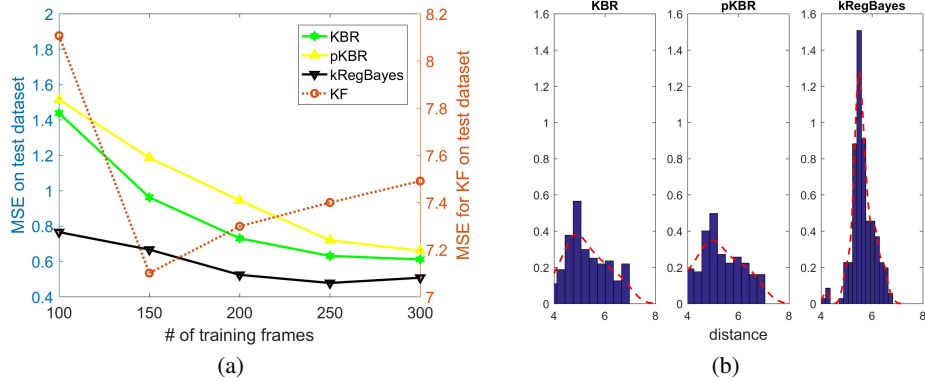

(a)

(b)

Figure 3: (a) MSEs for different algorithms (best view in color). Since KF performs much worse than kernel filters, we use a different scale and plot it on the right $y$-axis. (b) Probability histograms for the distance between each state and the scene center. All algorithms use 100 training data.

To provide supervision for kRegBayes, we uniformly generate 2000 data points $\{(\hat{x}_i, \hat{y}_t)\}_{i=1}^{2000}$ on the circle $r = 6$. Given the previous estimate $(\tilde{x}_t, \tilde{y}_t)$, we first compute $\hat{\theta}_t = \arctan(\hat{y}_t/\hat{x}_t)$ (where the value $\hat{\theta}_t$ is adapted according to the quadrant of $(\hat{x}_t, \hat{y}_t)$) and estimate $(\breve{x}_{t+1}, \breve{y}_{t+1}) = (\cos(\hat{\theta}_t + 0.4), \sin(\hat{\theta}_t + 0.4))$. Next, we find the nearest point to $(\breve{x}_{t+1}, \breve{y}_{t+1})$ in the supervision set $(\tilde{x}_k, \tilde{y}_k)$ and add the regularization $\mu_T \|\mu(\mathcal{I}_{t+1}) - \phi(\tilde{x}_k, \tilde{y}_k)\|$ to the posterior embedding, where $\mathcal{I}_{t+1}$ denotes the $(t + 1)$-th image.

We vary the size of training dataset from 100 to 300 and report the results of KBR, pKBR, kRegBayes and KF on 200 test images in Fig. 3. KF performs much worse than all three kernel filters due to the extreme non-linearity. The result of pKBR is a little worse than that of KBR, but the gap decreases as the training dataset becomes larger. kRegBayes always performs the best. Note that the advantage becomes less obvious as more data come. This is because kernel methods can learn the distance relation better with more data, and posterior regularization tends to be more useful when data are not abundant and domain knowledge matters. Furthermore, Fig. 3(b) shows that the posterior regularization helps the distances to concentrate.

## 6    Conclusions

We propose an optimizational framework for kernel Bayesian inference. With thresholding regularization, the minimizer of the framework is shown to be a reasonable estimator of the posterior kernel embedding. In addition, we propose a posterior regularized kernel Bayesian inference framework called kRegBayes. These frameworks are applied to non-linear state-space filtering tasks and the results of different algorithms are compared extensively.

## Acknowledgements

We thank all the anonymous reviewers for valuable suggestions. The work was supported by the National Basic Research Program (973 Program) of China (No. 2013CB329403), National NSF of China Projects (Nos. 61620106010, 61322308, 61332007), the Youth Top-notch Talent Support Program, and Tsinghua Initiative Scientific Research Program (No. 20141080934).

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
