[Supplementary Material]

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

# A  Appendix

## A.1  Kernel filtering

We first review how to use kernel techniques to do state-space filtering [5]. Assume that a sample $(y_1, x_1, \cdots, y_{T+1}, x_{T+1})$ is given, in which $y_i \in \mathcal{Y}$ is the state and $x_i \in \mathcal{X}$ is the corresponding observation. The transition and observation probabilities are estimated empirically in a nonparametric way:

$$\widehat{\mathcal{C}}_{YY_+} = \frac{1}{T} \sum_{i=1}^{T} \psi(y_i) \otimes \psi(y_{i+1}), \quad \widehat{\mathcal{C}}_{YX} = \frac{1}{T} \sum_{i=1}^{T} \psi(y_i) \otimes \phi(x_i).$$

The filtering task is composed of two steps. The first step is to predict the next state based on current state, *i.e.*, $p(Y_{t+1} \mid X_1, \cdots, X_t) = \int p(Y_{t+1} \mid Y_t) p(Y_t \mid X_1, \cdots, X_t) dY_t$. The second step is to update the state based on a new observation $x_{t+1}$ via Bayes' rule, *i.e.*, $p(Y_{t+1} \mid X_1, \cdots, X_{t+1}) \propto p(Y_{t+1} \mid X_1, \cdots, X_t) p(X_{t+1} \mid Y_{t+1})$. Following these two steps, we can obtain a recursive kernel update formula under different assumptions of the forms of kernel embedding $\widehat{m}_{y_t \mid x_1, \cdots, x_t}$.

For kernel embeddings without posterior regularization, we suppose $\widehat{m}_{y_t \mid x_1, \cdots, x_t} = \sum_{i=1}^{T} \alpha_i^{(t)} \psi(y_i)$. According to Thm. 1, the prediction step is realized by $\widehat{m}_{y_{t+1} \mid x_1, \cdots, x_t} = \widehat{\mathcal{C}}_{Y_+ Y}(\widehat{\mathcal{C}}_{YY} + \lambda_T I)^{-1} \widehat{m}_{y_t \mid x_1, \cdots, x_t} = \Psi_+ (G_Y + T\lambda_T I)^{-1} G_Y \boldsymbol{\alpha}^{(t)}$, where $\Psi_+ = (\psi(y_2), \cdots, \psi(y_{T+1}))$, $G_Y$ is the Gram matrix of $\{y_1, \cdots, y_T\}$ and $\boldsymbol{\alpha}^{(t)}$ is the vector of coefficients. The update step can be realized by invoking Prop. 2, *i.e.*, $\widehat{m}_{y_{t+1} \mid x_1, \cdots, x_{t+1}} = \Psi (K_X + \delta_T \Lambda^+)^{-1} K_{:x_{t+1}}$, where $K_X$ is the Gram matrix for $(x_1, \cdots, x_t)$, $\Lambda^+ = \text{diag}(1/\boldsymbol{\beta}^+)$ and $\boldsymbol{\beta} = (G_Y + T\lambda_T I)^{-1} G_{YY_+}(G_Y + T\lambda_T I)^{-1} G_Y \boldsymbol{\alpha}^{(t)}$, where $(G_{YY_+})_{ij} = k_{\mathcal{Y}}(y_i, y_{i+1})$. The update formula of $\boldsymbol{\alpha}^{(t+1)}$ can then be summarized as follows

$$\boldsymbol{\alpha}^{(t+1)} = (K_X + \delta_T \Lambda^+)^{-1} K_{:x_{t+1}}. \tag{13}$$

For kernel embeddings with posterior regularization, we suppose that for each step $t$, the regularization $\mu_T \|\mu(\tilde{x}_t) - \psi(\tilde{y}_t)\|$ is used, meaning that $p(Y_t | X_1, \cdots, X_t = \tilde{x}_t)$ is encouraged to concentrate around $\delta(Y_t = \tilde{y}_t)$. To obtain a recursive formula, we assume that $\widehat{m}_{y_t \mid x_1, \cdots, x_t} = \sum_{i=1}^{T} \alpha_i^{(t)} \psi(y_i) + \sum_{i=1}^{N} \tilde{\alpha}_i^{(t)} \psi(\tilde{y}_i)$, where $N$ is the number of supervision data points $(\tilde{x}_i, \tilde{y}_i)$. Following a similar logic except replacing Prop. 2 with Prop. 3, we get the update rule for $\boldsymbol{\alpha}^{(t+1)}$ and $\tilde{\boldsymbol{\alpha}}^{(t+1)}$

$$\boldsymbol{\gamma} = (K_X' + \delta_T \Lambda^\dagger)^{-1} K_{:x_{t+1}}' \tag{14}$$

$$\boldsymbol{\alpha}^{(t+1)} = \boldsymbol{\gamma}[1:m] \tag{15}$$

$$\tilde{\boldsymbol{\alpha}}^{(t+1)} = (0, \cdots, \boldsymbol{\gamma}[m+1], 0, \cdots)^{\mathsf{T}}, \tag{16}$$

where $\Lambda^\dagger = \text{diag}(1/\boldsymbol{\beta}^+, 1/\mu_T)$, $\boldsymbol{\beta} = (G_Y + T\lambda_T I)^{-1} G_{YY+}(G_Y Y + T\lambda_T I)^{-1}(G_{YY} \boldsymbol{\alpha}^{(t)} + G_{Y\tilde{Y}} \tilde{\boldsymbol{\alpha}}^{(t)})$. $K_X'$ and $K_{:x_{t+1}}'$ are augmented Gram matrices, which incorporate $(\tilde{x}_i, \tilde{y}_i)$. The position of $\boldsymbol{\gamma}[m+1]$ in $\tilde{\boldsymbol{\alpha}}^{(t+1)}$ corresponds to the index of supervision $(\tilde{x}_k, \tilde{y}_k)$ at $t+1$ step in $\{(\tilde{x}_i, \tilde{y}_i)\}_{i=m+1}^{n}$.

To obtain $\boldsymbol{\alpha}^{(1)}$, we use conditional operators [4] to estimate $m_{y_1}$ without priors. We set $\boldsymbol{\alpha}^{(1)} = (K_X + T\lambda_T I)^{-1} K_{:x_1}$ for both types of kernel filtering and $\tilde{\boldsymbol{\alpha}}^{(1)} = \mathbf{0}$. To decode the state from kernel embeddings, we solve an optimization problem $\hat{y}_t = \arg\min_y \|m(x) - \psi(y)\|$, which can be computed using an iteration scheme as depicted in [4].

## A.2  Proofs

**Proposition 1.** *Suppose $(X, Y)$ is a random variable in $\mathcal{X} \times \mathcal{Y}$, where the prior for $Y$ is $\pi(Y)$ and the likelihood is $p(X \mid Y)$. Let $\mathcal{H}_{\mathcal{X}}$ be a RKHS with kernel $k_{\mathcal{X}}$ and feature map $\phi(x)$, $\mathcal{H}_{\mathcal{Y}}$ be a RKHS with kernel $k_{\mathcal{Y}}$ and feature map $\psi(y)$, $\phi(x, y)$ be the feature map of $\mathcal{H}_{\mathcal{X}} \otimes \mathcal{H}_{\mathcal{Y}}$, $\widehat{\pi}_Y = \sum_{i=1}^{l} \tilde{\alpha}_i \psi(\tilde{y}_i)$ be an estimator for $\pi_Y$ and $\{(x_i, y_i)\}_{i=1}^{n}$ be a sample representing $p(X \mid Y)$. Under the assumption that $f(x, y) = \|\psi(y) - \mu(x)\|_{\mathcal{H}_{\mathcal{Y}}}^2 \in \mathcal{H}_{\mathcal{X}} \otimes \mathcal{H}_{\mathcal{Y}}$, we have*

$$\widehat{\mathcal{E}}_s[\mu] = \sum_{i=1}^{n} \beta_i \|\psi(y_i) - \mu(x_i)\|_{\mathcal{H}_{\mathcal{Y}}}^2, \tag{17}$$

where $\boldsymbol{\beta} = (\beta_1, \cdots, \beta_n)^\intercal$ is given by $\boldsymbol{\beta} = (G_Y + n\lambda I)^{-1}\tilde{G}_Y\tilde{\boldsymbol{\alpha}}$, where $(G_Y)_{ij} = k_{\mathcal{Y}}(y_i, y_j)$, $(\tilde{G}_Y)_{ij} = k_{\mathcal{Y}}(y_i, \tilde{y}_j)$, and $\tilde{\boldsymbol{\alpha}} = (\tilde{\alpha}_1, \cdots, \tilde{\alpha}_l)^\intercal$.

*Proof.* The reasoning is similar to [5], Prop. 5. We only need to show that $\widehat{\mu}_{(X,Y)} = \Phi_{X,Y}\boldsymbol{\beta} = \Phi_{X,Y}(G_Y + n\lambda I)^{-1}\tilde{G}_Y\tilde{\boldsymbol{\alpha}}$, where $\Phi_{X,Y} = (\phi(x_1, y_1), \cdots, \phi(x_n, y_n))$. Recall that $\widehat{\mu}_{(X,Y)} = \widehat{\mathcal{C}}_{(X,Y)Y}(\widehat{\mathcal{C}}_{YY} + \lambda I)^{-1}\widehat{\pi}_Y$. Let $h = (\widehat{\mathcal{C}}_{YY} + \lambda I)^{-1}\widehat{\pi}_Y$ and decompose it as $h = \sum_{i=1}^n a_i\psi(y_i) + h_\perp$, where $h_\perp$ is perpendicular to $\mathrm{span}\{\psi(y_1), \cdots, \psi(y_n)\}$. Expanding $(\widehat{\mathcal{C}}_{YY} + \lambda I)h = \widehat{\pi}_Y$, we obtain

$$\frac{1}{n}\sum_{i,j\leq n} a_i k_{\mathcal{Y}}(y_i, y_j)\psi(y_j) + \lambda(\sum_{i\leq n} a_i\psi(y_i) + h_\perp) = \sum_{i\leq l}\tilde{\alpha}_i\psi(\tilde{y}_i). \tag{18}$$

Multiplying both sides with $\psi(y_k)|_{k=1}^n$, we get $\frac{1}{n}G_Y^2\boldsymbol{a} + \lambda G_Y\boldsymbol{a} = \tilde{G}_Y\tilde{\boldsymbol{\alpha}}$. Therefore $\widehat{\mu}_{(X,Y)}$ can be written as $\widehat{\mu}_{(X,Y)} = \frac{1}{n}[\sum_{i\leq n}\phi(x_i, y_i)\otimes\psi(y_i)]h = \frac{1}{n}\Phi_{X,Y}G_Y\boldsymbol{a} = \Phi_{X,Y}(G_Y + n\lambda I)^{-1}\tilde{G}_Y\tilde{\boldsymbol{\alpha}}$. $\qquad\square$

**Proposition 2.** *Without loss of generality, we assume that $\beta_i^+ \neq 0$ for all $1 \leq i \leq n$. Let $\mu \in \mathcal{H}_K$ and choose the kernel of $\mathcal{H}_K$ to be $K(x_i, x_j) = k_{\mathcal{X}}(x_i, x_j)\mathcal{I}$, where $\mathcal{I} : \mathcal{H}_K \to \mathcal{H}_K$ is an identity map. Then*

$$\widehat{\mu}_{\lambda,n}(x) = \Psi(K_X + \lambda_n\Lambda^+)^{-1}K_{:x}, \tag{19}$$

*where $\Psi = (\psi(y_1), \cdots, \psi(y_n))$, $(K_X)_{ij} = k_{\mathcal{X}}(x_i, x_j)$, $\Lambda^+ = \mathrm{diag}(1/\beta_1^+, \cdots, 1/\beta_n^+)$, $K_{:x} = (k_{\mathcal{X}}(x, x_1), \cdots, k_{\mathcal{X}}(x, x_n))^\intercal$ and $\lambda_n$ is a positive regularization constant.*

*Proof.* If $\beta_i^+ = 0$ for any $i$, we can discard the data point $(x_i, y_i)$ without affecting results. Let $\mu = \mu_0 + g$, where $\mu_0 = \sum_{i=1}^n K_{x_i}c_i$. Plugging $\mu = \mu_0 + g$ into $\widehat{\mathcal{E}}_{\lambda,n}[\mu]$ and expand, we obtain $\widehat{\mathcal{E}}_{\lambda,n}[\mu] = \sum_{i=1}^n \beta_i^+ \|\psi(y_i) - \mu_0(x_i)\|^2 + \lambda_n \|\mu_0\|^2 + \sum_{i=1}^n \beta_i^+ \|g(x_i)\|^2 + \lambda_n \|g\|^2 + 2\lambda_n\langle\mu_0, g\rangle - 2\sum_{i=1}^n \beta_i^+\langle g(x_i), \psi(y_i) - \mu_0(x_i)\rangle$.

We conjecture that $\psi(y_i) - \sum_{j=1}^n k_{\mathcal{X}}(x_i, x_j)c_j = \frac{\lambda_n}{\beta_i^+}c_i$, for all $1 \leq i \leq n$. Actually, substituting these equations into $\widehat{\mathcal{E}}_{\lambda,n}[\mu]$ gives the relation $\lambda_n\langle\mu_0, g\rangle - \sum_{i=1}^n \beta_i^+\langle g(x_i), \psi(y_i) - \mu_0(x_i)\rangle = 0$. As a result, $\widehat{\mathcal{E}}_{\lambda,n}[\mu] = \widehat{\mathcal{E}}_{\lambda,n}[\mu_0] + \sum_{i=1}^n \beta_i^+ \|g(x_i)\|^2 + \lambda_n \|g\|^2 \geq \widehat{\mathcal{E}}_{\lambda,n}[\mu_0]$, which means that $\mu_0 = \sum_{i=1}^n K_{x_i}c_i$ with $c_i$ satisfying the conjectured equations is the solution. The equation $\psi(y_i) - \sum_{j=1}^n k_{\mathcal{X}}(x_i, x_j)c_j = \frac{\lambda_n}{\beta_i^+}c_i$ implies that $(K_X + \lambda_n\Lambda^+)c = \Psi$ and $\mu_0(x) = \sum_{i=1}^n k_{\mathcal{X}}(x, x_i)c_i = \Psi(K_X + \lambda_n\Lambda^+)^{-1}K_{:x}$. $\qquad\square$

**Theorem 6.** *Assume that $|\mathcal{X}\times\mathcal{Y}| < \infty$, $k$ is strictly positive definite with $\sup_{(x,y)} k((x,y),(x,y)) < \kappa$ and $f(x,y) = \|\psi(y) - \mu(x)\|_{\mathcal{H}_{\mathcal{Y}}}^2 \in \mathcal{H}_{\mathcal{X}}\otimes\mathcal{H}_{\mathcal{Y}}$. With the conditions in Thm. 2, we assert that $\widehat{\mu}_{(X,Y)}^+$ is a consistent estimator of $\mu_{(X,Y)}$ and $\left|\widehat{\mathcal{E}}_s^+[\mu] - \mathcal{E}_s[\mu]\right| \to 0$ in probability as $n \to \infty$.*

*Proof.* We only need to show that $\widehat{\mu}_{(X,Y)}^+ := \sum_{i=1}^n \beta_i^+\phi(x_i)\otimes\psi(y_i)$ converges to $\mu_{(X,Y)}$ in probability as $n \to \infty$, since $\left|\widehat{\mathcal{E}}_s^+[\mu] - \mathcal{E}_s[\mu]\right| = |\langle f, \widehat{\mu}_{(X,Y)}^+ - \mu_{(X,Y)}\rangle| \leq \|f\|\left\|\widehat{\mu}_{(X,Y)}^+ - \mu_{(X,Y)}\right\|$. From Thm. 2 we know that $\widehat{\mu}_{(X,Y)}$ converges to $\mu_{(X,Y)}$ in probability, hence it is sufficient to show that $\widehat{\mu}_{(X,Y)}^+$ converges to $\widehat{\mu}_{(X,Y)}$ in RKHS norm as $n \to \infty$.

Let $|\mathcal{X}\times\mathcal{Y}| = M$. Without losing generality, we assume $\mathcal{X}\times\mathcal{Y} = \{(x_1, y_1), \cdots, (x_M, y_M)\}$ and $\{(x_1, y_1), \cdots, (x_n, y_n)\}$ is a sample representing $p(X \mid Y)$. According to Theorem 4 in [24], $k$ is strictly positive definite on a finite set implies that $\mathcal{H}_{\mathcal{X}}\otimes\mathcal{H}_{\mathcal{Y}}$ consists of all bounded functions on $\mathcal{X}\times\mathcal{Y}$. In particular, $\mathcal{H}_{\mathcal{X}}\otimes\mathcal{H}_{\mathcal{Y}}$ contains the function

$$g(x_i, y_i) = \begin{cases} 1, & \beta_i < 0 \\ 0, & \text{otherwise.} \end{cases} \tag{20}$$

We denote $b := \max_g \|g\|_{\mathcal{H}_{\mathcal{X}}\otimes\mathcal{H}_{\mathcal{Y}}} = \max_{\mathbf{g}} \mathbf{g}^\intercal K^{-1}\mathbf{g}$ for all possibilities of $\boldsymbol{\beta}$. Here $\mathbf{g}$ represents the point evaluations of $g$ on $\{(x_i, y_i)\}_{i=1}^M$ and $K_{ij}|_{1\leq i,j\leq M} = k((x_i, y_i), (x_j, y_j))$. Note that $g(x, y)$ is non-negative, thus $\mathbb{E}[g(X, Y)] = \langle g, \mu_{(X,Y)}\rangle \geq 0$. For sufficiently large $n$, $|\langle g, \widehat{\mu}_{(X,Y)} - $

$\mu_{(X,Y)}\rangle| \le \|g\| \, \|\widehat{\mu}_{(X,Y)} - \mu_{(X,Y)}\| \le \epsilon b$ in arbitrarily high probability. In this case $\langle g, \widehat{\mu}_{(X,Y)}\rangle = -\sum_{i=1}^n \beta_i^- \ge -\epsilon b$, where $\beta_i^- = -\min(0, \beta_i)$, and $\left\|\widehat{\mu}_{(X,Y)}^+ - \widehat{\mu}_{(X,Y)}\right\| = \left\|\sum_{i=1}^n \beta_i^- \phi(x_i, y_i)\right\| = \sqrt{\sum_{i,j} \beta_i^- \beta_j^- k((x_i, y_i), (x_j, y_j))} \le \sqrt{\kappa} \sum_{i=1} \beta_i^- \le \epsilon b \sqrt{\kappa}$. The inequalities can now be linked and the theorem proved. $\qquad\square$

**Theorem 7.** *Assume that $|\mathcal{X} \times \mathcal{Y}| < \infty$, $k$ is strictly positive definite with $\sup_{(x,y)} k((x,y),(x,y)) < \kappa$, we assert $\sum_{i=1}^n \beta_i^+ \to 1$ in probability as $n \to \infty$.*

*Proof.* The proof follows a similar reasoning to that in Thm. 6. Let $|\mathcal{X} \times \mathcal{Y}| = M$ and $\{(x_1, y_1), \cdots, (x_n, y_n)\}$ be a sample representing $p(X \mid Y)$. According to Theorem 4 in [24], $k$ is strictly positive definite on a finite set implies that $\mathcal{H}_\mathcal{X} \otimes \mathcal{H}_\mathcal{Y}$ consists of all bounded functions on $\mathcal{X} \times \mathcal{Y}$. In particular, $\mathcal{H}_\mathcal{X} \otimes \mathcal{H}_\mathcal{Y}$ contains the function $f(x,y) \equiv 1$. From Thm. 6 we know that $\widehat{\mu}^+ = \sum_{i=1}^n \beta_i^+ \phi(x_i) \otimes \psi(y_i) \to \mu$ in probability. Therefore, $|\sum_{i=1}^n \beta_i^+ - 1| = |\langle f, \widehat{\mu}_{(X,Y)}^+ - \mu_{(X,Y)}\rangle| \le \|f\| \left\|\widehat{\mu}_{(X,Y)}^+ - \mu_{(X,Y)}\right\| \to 0$ in probability. $\qquad\square$

Since $\beta_i$'s do not depend on $X_1, \cdots, X_n$, we have the following corollary:

**Corollary 1.** *Assume that $|\mathcal{Y}| < \infty$, $k$ is strictly positive definite with $\sup_{(x,y)} k((x,y),(x,y)) < \kappa$, we assert $\sum_{i=1}^n \beta_i^+ \to 1$ in probability as $n \to \infty$.*

Next, we will relax the finite space condition on $\mathcal{X} \times \mathcal{Y}$ in Thm. 6. To this end, we introduce the following convenient concept of $\epsilon$-partition.

**Definition 1** ($\epsilon$-partition). *An $\epsilon$-partition of a metric space $\mathcal{X}$ is a partition whose elements are all within $\epsilon$-balls of $\mathcal{X}$.*

Since a compact space is totally bounded, we have the more general result.

**Theorem 3.** *Assume that $\mathcal{X}$ is compact and $|\mathcal{Y}| < \infty$, $k$ is a strictly positive definite continuous kernel with $\sup_{(x,y)} k((x,y),(x,y)) < \kappa$ and $f(x,y) = \|\psi(y) - \mu(x)\|_{\mathcal{H}_\mathcal{Y}}^2 \in \mathcal{H}_\mathcal{X} \otimes \mathcal{H}_\mathcal{Y}$. With the conditions in Thm. 2, we assert that $\widehat{\mu}_{(X,Y)}^+$ is a consistent estimator of $\mu_{(X,Y)}$ and $\left|\widehat{\mathcal{E}}_s^+[\mu] - \mathcal{E}_s[\mu]\right| \to 0$ in probability as $n \to \infty$.*

*Proof.* From the condition that $\phi(x,y)$ is continuous on the compact space $\mathcal{X} \times \mathcal{Y}$, we know $\phi(x,y)$ and $\phi(x)$ are uniformly continuous.

For any probability measure $p$ and $\epsilon$-partition of $\mathcal{X}$, we can construct a new discretized probability measure in the following way. Suppose the $\epsilon$-partition is $\{B_1^\epsilon, B_2^\epsilon, \cdots\}$, we identify each set $B_i^\epsilon$ with a representative element $x_i^c \in B_i^\epsilon$. The resulting probability measure is denoted as $p^\epsilon$ and satisfies $p^\epsilon(A) = \sum_{x_i^c \in A} p(B_i^\epsilon)$. We also define the discretization $x_i^\epsilon$ of $x_i$ to be $x_i^\epsilon = x_j^c$ if $x_i \in B_j$. Let the kernel embedding of $p$ be $\mu$ and $p^\epsilon$ be $\mu^\epsilon$. Suppose $\forall \delta > 0$, $\exists \epsilon > 0$ such that $\|x_1 - x_2\| \le \epsilon$ implies $\|\phi(x_1) - \phi(x_2)\|_{\mathcal{H}_\mathcal{X}} \le \delta$. We assert that $\|\mu - \mu^\epsilon\| \le \delta$. To prove this, we observe that an i.i.d. sample $\{x_1, \cdots, x_n\}$ from $p$ is also an i.i.d. sample of $p^\epsilon$ if we replace $x_i$ with $x_i^\epsilon$. Since the estimator $\widehat{\mu} = \frac{1}{n}\sum_{i=1}^n \phi(x_i)$ is a consistent estimator of $\mu$, we know that $\widehat{\mu^\epsilon} = \frac{1}{n}\sum_{i=1}^n \phi(x_i^\epsilon)$ is also consistent. Via consistency, we have that with no less than any high probability $1 - \Delta$, for any $n > N(\Delta, \delta', \epsilon)$, $\|\widehat{\mu} - \mu\| \le \delta'$ and $\|\widehat{\mu^\epsilon} - \mu^\epsilon\| \le \delta'$ holds. Since $\|\widehat{\mu} - \widehat{\mu^\epsilon}\| \le \frac{1}{n}\sum_{i=1}^n \|\phi(x_i) - \phi(x_i^\epsilon)\|$ and $\|x_i - x_i^\epsilon\| \le \epsilon$, we have $\|\widehat{\mu} - \widehat{\mu^\epsilon}\| \le \delta$ from uniform continuity. Combining this with $\|\widehat{\mu} - \mu\| \le \delta'$ and $\|\widehat{\mu^\epsilon} - \mu^\epsilon\| \le \delta'$ we know $\|\mu - \mu^\epsilon\| \le \|\mu - \widehat{\mu}\| + \|\widehat{\mu} - \widehat{\mu^\epsilon}\| + \|\widehat{\mu^\epsilon} - \mu^\epsilon\| \le \delta + 2\delta'$ with high probability $1 - \Delta$. Note that $\|\mu - \mu^\epsilon\| \le \delta + 2\delta'$ is a deterministic event and holds for any $\delta' > 0$, we have $\|\mu - \mu^\epsilon\| \le \delta$.

Now we would like to discretize $X$ for $\mu_{(X,Y)}$. For any $\epsilon > 0$, we have $\widehat{\mu}_{(X,Y)} = \sum_{i=1}^n \beta_i \phi(x_i, y_i) \to \mu_{(X,Y)}$ in probability and $\widehat{\mu^\epsilon}_{(X,Y)} = \sum_{i=1}^n \beta_i^\epsilon \phi(x_i^\epsilon, y_i) \to \mu_{(X,Y)}^\epsilon$ in probability. Since $\beta_i$ depends only on $y_1, \cdots, y_n$, we have $\beta_i = \beta_i^\epsilon$. From the last paragraph we suppose that

$\epsilon$ is chosen such that $\forall \|(x_i^\epsilon, y_i) - (x_i, y_i)\| \le \epsilon$, $\|\phi(x_i^\epsilon, y_i) - \phi(x_i, y_i)\| \le \delta$. Note that

$$
\left\| \sum_{i=1}^{n} \beta_i^+ \phi(x_i, y_i) - \mu_{(X,Y)} \right\| \le \left\| \sum_{i=1}^{n} \beta_i^+ \phi(x_i, y_i) - \sum_{i=1}^{n} \beta_i^+ \phi(x_i^\epsilon, y_i) \right\|
$$

$$
+ \left\| \sum_{i=1}^{n} \beta_i^+ \phi(x_i^\epsilon, y_i) - \sum_{i=1}^{n} \beta_i \phi(x_i^\epsilon, y_i) \right\|
$$

$$
+ \left\| \sum_{i=1}^{n} \beta_i \phi(x_i^\epsilon, y_i) - \mu_{(X,Y)}^\epsilon \right\| + \left\| \mu_{(X,Y)}^\epsilon - \mu_{(X,Y)} \right\|
$$

$$
\le \delta \sum_{i=1}^{n} \beta_i^+ + \delta + \left\| \sum_{i=1}^{n} \beta_i^+ \phi(x_i^\epsilon, y_i) - \sum_{i=1}^{n} \beta_i \phi(x_i^\epsilon, y_i) \right\|
$$

$$
+ \left\| \sum_{i=1}^{n} \beta_i \phi(x_i^\epsilon, y_i) - \mu_{(X,Y)}^\epsilon \right\|.
$$

From Corollary 1, Thm. 6 and the consistency of $\sum_{i=1}^{n} \beta_i \phi(x_i^\epsilon, y_i)$ we see $\left\| \sum_{i=1}^{n} \beta_i^+ \phi(x_i, y_i) - \mu_{(X,Y)} \right\|$ can be arbitrarily small with arbitrarily high probability. This proves the consistency of $\sum_{i=1}^{n} \beta_i^+ \phi(x_i, y_i)$. $\qquad\square$

**Corollary 2.** *Assume that $\mathcal{X}$ is compact and $|\mathcal{Y}| < \infty$, $k$ is a bounded strictly positive definite continuous kernel, $k_{\mathcal{X}}$ is a bounded kernel with $\sup_x k_{\mathcal{X}}(x,x) \le \kappa_{\mathcal{X}}$, we assert that $\widehat{\mu}_X^+ = \sum_{i=1}^{n} \beta_i^+ \phi(x_i)$ is a consistent estimator of $\mu_X$, i.e., the kernel embedding of the marginal distribution on $X$.*

**Theorem 8.** *Let $\mathcal{B}_1$, $\mathcal{B}_2$ be Banach spaces. For any linear operator $\mathcal{A} : \mathcal{B}_1 \to \mathcal{B}_2$, we assert that there exists a subset $\mathcal{F} \subseteq \mathcal{B}_1$ such that $\mathcal{F}$ is dense in $\mathcal{B}_1$ and $\|\mathcal{A}f\|_{\mathcal{B}_2} \le N \|f\|_{\mathcal{B}_1}$ for some constant $N$ and any $f \in \mathcal{F}$.*

*Proof.* Let $M_k$ be the set of $f \in \mathcal{B}_1$ satisfying $\|\mathcal{A}f\|_{\mathcal{B}_2} \le k \|f\|_{\mathcal{B}_1}$. Clearly we have $\mathcal{B}_1 = \bigcup_{k=1}^{\infty} M_k$. Since $\mathcal{B}_1$ is complete, we can invoke Baire category theorem to conclude that there exists an integer $n$ such that $M_n$ is dense in some sphere $S_0 \subseteq \mathcal{B}_1$. Consider the spherical shell $P$ in $S_0$ consisting of the points $z$ for which

$$
\beta < \|z - y_0\| < \alpha,
$$

where $0 < \beta < \alpha$, $y_0 \in M_n$. Next, translate the spherical shell $P$ so that its center coincides with the origin of coordinates to obtain spherical shell $P_0$. We now show that there is some set $M_N$ dense in $P_0$. For every $z \in M_n \cap P$, we have

$$
\|\mathcal{A}(z - y_0)\|_{\mathcal{B}_2} \le \|\mathcal{A}z\|_{\mathcal{B}_2} + \|\mathcal{A}y_0\|_{\mathcal{B}_2} \le n(\|z\|_{\mathcal{B}_1} + \|y_0\|_{\mathcal{B}_1}) \le n(\|z - y_0\|_{\mathcal{B}_1} + 2\|y_0\|_{\mathcal{B}_1})
$$

$$
= n\|z - y_0\|_{\mathcal{B}_1} [1 + 2\|y_0\|_{\mathcal{B}_1} / \|z - y_0\|_{\mathcal{B}_1}] \le n\|z - y_0\|_{\mathcal{B}_1} [1 + 2\|y_0\|_{\mathcal{B}_1} / \beta].
$$

Let $N = n(1 + 2\|y_0\|_{\mathcal{B}_1} / \beta)$, we have $z - y_0 \in M_N$. Since $z - y_0 \in M_N$ is obtained from $z \in M_n$ and $M_n$ is dense in $P$, it is easy to see that $M_N$ is dense in $P_0$. For any $y \in \mathcal{B}_1$ except $\|y\|_{\mathcal{B}_1} = 0$, it is always possible to choose $\lambda$ so that $\beta < \|\lambda y\| < \alpha$ and we can construct a sequence $y_k \in M_N$ that converges to $\lambda y$. This means there exists a sequence $(1/\lambda)y_k$ converging to $y$. By virtue of $(1/\lambda)y_k \in M_N$ and $0 \in M_N$, we conclude $M_N$ is dense in $\mathcal{B}_1$. $\qquad\square$

**Theorem 9.** *Let $(\Omega, \mathcal{F}, P)$ be a probability space and $\xi$ be a random variable on $\Omega$ taking values in a Hilbert space $\mathcal{K}$. Define $\mathcal{A} : f \in \mathcal{K} \mapsto \langle f, \xi(\cdot) \rangle \in \mathcal{H}$, where $\mathcal{H}$ is a RKHS with feature maps $\phi(\omega)$. Let $\mu$ be a kernel embedding for $P^\pi$ and $\widehat{\mu} = \sum_{i=1}^{n} \beta_i^+ \phi(\omega_i)$ be a consistent estimator of $\mu$. Assume $\sum_{i=1}^{n} \beta_i^+ \to 1$ in probability and there are two positive constants $H$ and $\sigma$ such that $\|\xi(\omega)\|_{\mathcal{K}} \le \frac{H}{2}$ a.s. and $\mathbb{E}_{P^\pi}[\|\xi\|_{\mathcal{K}}^2] \le \sigma^2$. Then for any $\epsilon > 0$,*

$$
\lim_{n \to 0} P^l \left[ (\omega_1, \cdots, \omega_l) \in \Omega^l \mid \left\| \sum_{i=1}^{n} \beta_i^+ \xi(w_i) - \mathbb{E}_{P^\pi}[\xi] \right\|_{\mathcal{K}} > \epsilon \right] = 0 \tag{21}
$$

*Proof.* From the consistency of $\widehat{\mu}$, we know for every $\epsilon_1$, there exists $N_{\epsilon_1}(\delta_1)$ such that $\forall n > N_{\epsilon_1}(\delta_1)$, $\|\widehat{\mu} - \mu\|_{\mathcal{H}} < \epsilon_1$ with probability no less than $1 - \delta_1$. Similarly, for every $\epsilon_2$, there exists $N_{\epsilon_2}(\delta_2)$ such that $\forall n > N_{\epsilon_2}(\delta_2)$, $\left|\sum_{i=1}^n \beta_i^+ - 1\right| < \epsilon_2$ with probability no less than $1 - \delta_2$. Furthermore, with probability no less than $1 - \delta_2$, $\left\|\sum_{i=1}^n \beta_i^+ \xi(w_i) - \mathbb{E}_{P^\pi}[\xi]\right\|_{\mathcal{K}} \le \sum_{i=1}^n \beta_i^+ \|\xi(w_i)\|_{\mathcal{K}} + \|\mathbb{E}_{P^\pi}[\xi]\|_{\mathcal{K}} \le (1 + \epsilon_2) \|\xi(\omega)\|_{\mathcal{K}} + \mathbb{E}_{P^\pi}[\|\xi\|] \le \frac{H(1+\epsilon_2)}{2} + \sqrt{\mathbb{E}_{P^\pi}[\|\xi\|_{\mathcal{K}}^2]} = \frac{H(1+\epsilon_2)}{2} + \sigma$, where the last two inequalities follow from Jensen's inequality.

Let $f = \sum_{i=1}^n \beta_i^+ \xi(w_i) - \mathbb{E}_{P^\pi}[\xi]$ and clearly $\|f\|_{\mathcal{K}} \le \frac{H(1+\epsilon_2)}{2} + \sigma$. Consider $\Delta_f := \sum_{i=1}^n \beta_i^+ \langle f, \xi(\omega_i)\rangle - \langle f, \mathbb{E}_{P^\pi}[\xi]\rangle = \sum_{i=1}^n \beta_i^+ [\mathcal{A}f](\omega_i) - \mathbb{E}_{P^\pi}[\mathcal{A}f] = \langle \widehat{\mu} - \mu, \mathcal{A}f\rangle$. In virtue of Thm. 8, for any $\epsilon_3$, there exists an element $g \in \mathcal{K}$ and constant $N$ (only depends on $\mathcal{A}$) such that $\|g - f\|_{\mathcal{K}} < \epsilon_3$ and $\|\mathcal{A}g\|_{\mathcal{H}} \le N\|g\|_{\mathcal{K}}$. Similarly define $\Delta_g := \sum_{i=1}^n \beta_i^+ \langle g, \xi(\omega_i)\rangle - \langle g, \mathbb{E}_{P^\pi}[\xi]\rangle = \langle \widehat{\mu} - \mu, \mathcal{A}g\rangle$. It is easy to see that $|\Delta_g - \Delta_f| \le (1+\epsilon_2)\epsilon_3 \|\xi(\omega)\|_{\mathcal{K}} + \epsilon_3 \|\mathbb{E}_{P^\pi}[\xi]\|_{\mathcal{K}} \le \frac{H\epsilon_3(1+\epsilon_2)}{2} + \epsilon_3\sigma$ and $\Delta_g = \langle \widehat{\mu} - \mu, \mathcal{A}g\rangle \le \epsilon_1 N\|g\|_{\mathcal{K}} \le \epsilon_1 N(\epsilon_3 + \|f\|_{\mathcal{K}}) \le \epsilon_1 N(\epsilon_3 + \sigma + \frac{H(1+\epsilon_2)}{2})$ with probability no less than $1 - \delta_1 - \delta_2$. Hence $\left\|\sum_{i=1}^n \beta_i^+ \xi(w_i) - \mathbb{E}_{P^\pi}[\xi]\right\|_{\mathcal{K}} = \sqrt{|\Delta_f|} \le \sqrt{\epsilon_1 N(\epsilon_3 + \sigma + \frac{H(1+\epsilon_2)}{2}) + \frac{H\epsilon_3(1+\epsilon_2)}{2} + \epsilon_3\sigma}$ with probability no less than $1 - \delta_1 - \delta_2$ for all $n > \max(N_{\epsilon_1}(\delta_1), N_{\epsilon_2}(\delta_2))$. The theorem now gets proved. $\square$

The proof of Thm. 5 is based on the proof of Thm. 5 in [20], with more assumptions and different concentration results. For convenience, we borrow some notations in their paper and refer the readers to [20] for definitions. We suggest the readers to be familiar with [20] because we modify and skip some details of the proofs to make the reasoning clearer.

Let $\mathcal{X}, \mathcal{Y}$ be Polish spaces, $\mathcal{H}_\mathcal{Y}$ be a separable Hilbert space, $\mathcal{Z} = \mathcal{X} \times \mathcal{Y}$, $\mathcal{H}_K$ be a real Hilbert space of functions $\mu : \mathcal{X} \to \mathcal{H}_\mathcal{Y}$ satisfying $\mu(x) = K_x^* \mu$ where $K_x : \mathcal{H}_\mathcal{Y} \to \mathcal{H}_K$ is the bounded operator $K_x v = K(\cdot, x)v, \quad v \in \mathcal{H}_\mathcal{Y}$. Moreover, let $T_x = K_x K_x^* \in \mathcal{L}_2(\mathcal{H}_K)$ be a positive Hilbert-Schmidt operator.

Let $\rho$ be a probability measure on $\mathcal{Z}$ and $\rho_X$ denotes the marginal distribution of $\rho$ on $\mathcal{X}$. We suppose that $\rho = p(X \mid Y)\pi(Y)$ and thus it incorporates the information of the prior. In contrast, we are given a sample $\mathbf{z} = ((x_1, y_1), \cdots, (x_n, y_n))$ from another distribution on $\mathcal{Z}$ with the same $p(X \mid Y)$.

The optimization objective now becomes $\mathcal{E}_s[\mu] = \int_{\mathcal{Z}} \|\mu(x) - \phi(y)\|_{\mathcal{H}_\mathcal{Y}}^2 \, d\rho(x, y)$. Denote $T = \int_{\mathcal{X}} T_x d\rho_X(x)$, $T_\mathbf{x} = \sum_{i=1}^n \beta_i^+ T_{x_i}$, $\mu_{\mathcal{H}_K} = \arg\min_\mu \mathcal{E}_s[\mu]$, $\mu^\lambda = \mathcal{E}_s[\mu] + \lambda\|\mu\|_{\mathcal{H}_K}^2$ and $\mu_\mathbf{z}^\lambda = \widehat{\mathcal{E}}_{\lambda,n}[\mu]$. Additionally, let $A : \mathcal{H}_K \to L^2(\mathcal{Z}, \rho, \mathcal{H}_\mathcal{Y})$ be the linear operator $(Af)(x, y) = K_x^* f \quad \forall (x, y) \in \mathcal{Z}$ and $A_\mathbf{z} := A_{\rho = \sum_{i=1}^n \beta_i^+ \delta_{x_i}}$. Finally, let $\mathcal{A}(\lambda) = \left\|\mu^\lambda - \mu_{\mathcal{H}_K}\right\|_\rho^2 = \left\|\sqrt{T}(\mu^\lambda - \mu_{\mathcal{H}_K})\right\|$, $\mathcal{B}(\lambda) = \left\|\mu^\lambda - \mu_{\mathcal{H}_K}\right\|_{\mathcal{H}_K}^2$ and $\mathcal{N}(\lambda) = \mathrm{Tr}((T + \lambda)^{-1}T)$.

**Assumption 1.** *Let $\mathcal{A}_1 : f \in \mathcal{L}_2(\mathcal{H}_K) \mapsto \langle f, (T + \lambda)^{-1}T.\rangle \in \mathcal{H}_1$, $\mathcal{A}_2 : f \in \mathcal{L}(\mathcal{H}_K) \mapsto \langle f, T.(\mu^\lambda - \mu_{\mathcal{H}_K})\rangle \in \mathcal{H}_2$, $\mathcal{A}_3 : f \in \mathcal{H}_K \mapsto \langle f, (T + \lambda)^{-\frac{1}{2}} K_{\#1}(\psi(\#2) - \mu_{\mathcal{H}_K}(\#1))\rangle \in \mathcal{H}_3$, where $\#1$ and $\#2$ denote two arguments of the function. We assume that $\mathcal{H}_1 = \mathcal{H}_2 = \mathcal{H}_\mathcal{X}$, $\mathcal{H}_3 = \mathcal{H}_\mathcal{X} \otimes \mathcal{H}_\mathcal{Y}$.*

**Assumption 2.** *We assume that $\widehat{\mu}_{(X,Y)}^+ = \sum_{i=1}^n \beta_i^+ \phi(x_i) \otimes \psi(y_i)$ is a consistent estimator of $\mu_{(X,Y)}$ and $\widehat{\mu}_\mathcal{X}^+ = \sum_{i=1}^n \beta_i^+ \phi(x_i)$ is also consistent for the kernel embedding of the marginal distribution on $X$. Furthermore, we assume $\sum_{i=1}^n \beta_i^+ \xrightarrow{p} 1$. Note that as shown in Thm. 3, Thm. 7 and Corollary 2, this hypothesis holds when $\mathcal{X}$ is compact and $\mathcal{Y}$ is finite.*

**Theorem 10.** *With the above Assumption 1, Assumption 2 and Hypothesis 1, Hypothesis 2 in [20], we assert that if $\lambda_n$ decreases to 0,*

$$\mathcal{E}_s[\mu_\mathbf{z}^{\lambda_n}] - \mathcal{E}_s[\mu_{\mathcal{H}_K}] \to 0 \tag{22}$$

*in probability as $n \to \infty$.*

*Proof.* This proof is adapted from that of Thm. 5 in [20]. We split the proof to 3 steps.

**Step 1**: Given a training set $\mathbf{z} = (\mathbf{x}, \mathbf{y}) \in \mathcal{Z}^n$, Prop. 2 in [20] gives

$$\mathcal{E}_s[\mu_\mathbf{z}^\lambda] - \mathcal{E}_s[\mu_{\mathcal{H}_K}] = \left\|\sqrt{T}(\mu_\mathbf{z}^\lambda - \mu_{\mathcal{H}_K})\right\|_{\mathcal{H}_K}^2.$$

As usual,

$$\mu_{\mathbf{z}}^\lambda - \mu_{\mathcal{H}_K} = (\mu_{\mathbf{z}}^\lambda - \mu^\lambda) + (\mu^\lambda - \mu_{\mathcal{H}_K})$$

Another application of Prop. 2 in [20] gives

$$\begin{aligned}
\mu_{\mathbf{z}}^\lambda - \mu^\lambda &= (T_{\mathbf{x}} + \lambda)^{-1} A_{\mathbf{z}}^* \psi(\mathbf{y}) - (T + \lambda)^{-1} A^* \psi(y) \\
&= (T_{\mathbf{x}} + \lambda)^{-1}(A_{\mathbf{z}}^* \psi(\mathbf{y}) - T_{\mathbf{x}} \mu_{\mathcal{H}_K}) + (T_{\mathbf{x}} + \lambda)^{-1}(T - T_{\mathbf{x}})(\mu^\lambda - \mu_{\mathcal{H}_K}).
\end{aligned}$$

From $\|\mu_1 + \mu_2 + \mu_3\|_{\mathcal{H}_K}^2 \le 3(\|\mu_1\|_{\mathcal{H}_K}^2 + \|\mu_2\|_{\mathcal{H}_K}^2 + \|\mu_3\|_{\mathcal{H}_K}^2)$,

$$\mathcal{E}_s[\mu_{\mathbf{z}}^\lambda] - \mathcal{E}_s[\mu_{\mathcal{H}_K}] \le 3(\mathcal{A}(\lambda) + \mathcal{S}_1(\lambda, \mathbf{z}) + \mathcal{S}_2(\lambda, \mathbf{z})), \tag{23}$$

where

$$\mathcal{S}_1(\lambda, \mathbf{z}) = \left\| \sqrt{T}(T_{\mathbf{x}} + \lambda)^{-1}(A_{\mathbf{z}}^* \psi(\mathbf{y}) - T_{\mathbf{x}} \mu_{\mathcal{H}_K}) \right\|_{\mathcal{H}_K}^2$$

$$\mathcal{S}_2(\lambda, \mathbf{z}) = \left\| \sqrt{T}(T_{\mathbf{x}} + \lambda)^{-1}(T - T_{\mathbf{x}})(\mu^\lambda - \mu_{\mathcal{H}_K}) \right\|_{\mathcal{H}_K}^2.$$

**Step 2**: probabilistic bound on $\mathcal{S}_2(\lambda, \mathbf{z})$. First

$$\mathcal{S}_2(\lambda, \mathbf{z}) \le \left\| \sqrt{T}(T_{\mathbf{x}} + \lambda)^{-1} \right\|_{\mathcal{L}(\mathcal{H}_K)}^2 \left\| (T - T_{\mathbf{x}})(\mu^\lambda - \mu_{\mathcal{H}_K}) \right\|_{\mathcal{H}_K}^2. \tag{24}$$

**Step 2.1**: probabilistic bound on $\left\| \sqrt{T}(T_{\mathbf{x}} + \lambda)^{-1} \right\|_{\mathcal{L}(\mathcal{H}_K)}$. We introduce an auxiliary quantity

$$\Theta(\lambda, \mathbf{z}) = \left\| (T + \lambda)^{-1}(T - T_{\mathbf{x}}) \right\|_{\mathcal{L}(\mathcal{H}_K)}$$

and assume

$$\Theta(\lambda, \mathbf{z}) \le \frac{1}{2}.$$

Invoking the Neumann series,

$$\left\| \sqrt{T}(T_{\mathbf{x}} + \lambda)^{-1} \right\|_{\mathcal{L}(\mathcal{H}_K)} = \sqrt{T}(T + \lambda)^{-1} \sum_{n=0}^{\infty} ((T + \lambda)^{-1}(T - T_{\mathbf{x}}))^n$$

$$\le \left\| \sqrt{T}(T + \lambda)^{-1} \right\|_{\mathcal{L}_{\mathcal{H}_K}} \sum_{n=0}^{\infty} \Theta(\lambda, n)^n$$

$$\text{(By spectral theorem)} \quad \le \frac{1}{2\sqrt{\lambda}} \frac{1}{1 - \Theta(\lambda, \mathbf{z})} \le \frac{1}{\sqrt{\lambda}} \tag{25}$$

We now claim that $\Theta(\lambda, \mathbf{z}) \le \frac{1}{2}$ with high probability as $n \to \infty$. Let $\xi_1 : \mathcal{X} \to \mathcal{L}_2(\mathcal{H}_K)$ be the random variable

$$\xi_1(x) = (T + \lambda)^{-1} T_x.$$

By the same reasoning in the proof of Thm. 5 in [20], we have $\|\xi_1\|_{\mathcal{L}_2(\mathcal{H}_K)} \le \frac{\kappa}{\lambda} = \frac{H_1}{2}$ and $\mathbb{E}[\|\xi_1\|_{\mathcal{L}_2(\mathcal{H}_K)}^2] \le \frac{\kappa}{\lambda}\mathcal{N}(\lambda) = \sigma_1^2$. Our assumptions and Thm. 9 ensure that for any $\delta_1$ there exists $N_1(\delta_1)$ such that

$$\Theta(\lambda, \mathbf{z}) = \left\| (T + \lambda)^{-1} T_{\mathbf{x}} - (T + \lambda)^{-1} T \right\|_{\mathcal{L}_2(\mathcal{H}_K)} \le \frac{1}{2}$$

with probability greater than $1 - \delta_1$ as long as $n > N_1(\delta_1)$.

**Step 2.2**: probabilistic bound on $\left\| (T - T_{\mathbf{x}})(\mu^\lambda - \mu_{\mathcal{H}_K}) \right\|_{\mathcal{L}(\mathcal{H}_K)}$. Let $\xi_2 : \mathcal{X} \to \mathcal{H}_K$ be the random variable

$$\xi_2(x) = T_x(\mu^\lambda - \mu_{\mathcal{H}_K}).$$

By the same reasoning, we have $\|\xi_2(x)\|_{\mathcal{H}_K} \leq \kappa\sqrt{\mathcal{B}(\lambda)} = \frac{H_2}{2}$ and $\mathbb{E}[\|\xi_2\|_{\mathcal{H}_K}^2] \leq \kappa\mathcal{A}(\lambda) = \sigma_2^2$. Applying our assumptions and Thm. 9 we conclude that for any $\delta_2, \epsilon_2$ there exists $N_2(\delta_2, \epsilon_2)$ such that

$$\left\|(T - T_{\mathbf{x}})(\mu^\lambda - \mu_{\mathcal{H}_K})\right\|_{\mathcal{H}_K} \leq \epsilon_2 \tag{26}$$

with probability greater than $1 - \delta_2$ as long as $n > N_2(\delta_2, \epsilon_2)$.

**Step 3**: probabilistic bound on $\mathcal{S}_1(\lambda, \mathbf{z})$. As usual,

$$\mathcal{S}_1(\lambda, \mathbf{z}) \leq \left\|\sqrt{T}(T_{\mathbf{x}} + \lambda)^{-1}(T + \lambda)^{1/2}\right\|_{\mathcal{L}(\mathcal{H}_K)}^2 \left\|(T + \lambda)^{-1/2}(A_{\mathbf{z}}^*\psi(\mathbf{y}) - T_{\mathbf{x}}\mu_{\mathcal{H}_K})\right\|_{\mathcal{H}_K}^2.$$

**Step 3.1**: bound $\left\|\sqrt{T}(T_{\mathbf{x}} + \lambda)^{-1}(T + \lambda)^{1/2}\right\|_{\mathcal{L}(\mathcal{H}_K)}$. Let

$$\Omega(\lambda, \mathbf{z}) = \left\|(T + \lambda)^{1/2}(T - T_{\mathbf{x}})(T + \lambda)^{-1/2}\right\|_{\mathcal{L}(\mathcal{H}_K)}$$

and assume $\Omega(\lambda, \mathbf{z}) \leq \frac{1}{2}$. Clearly,

$$\left\|\sqrt{T}(T_{\mathbf{x}} + \lambda)^{-1}(T + \lambda)^{1/2}\right\|_{\mathcal{L}(\mathcal{H}_K)} \tag{27}$$

$$= \left\|\sqrt{T}(T + \lambda)^{-1/2}\{I - (T + \lambda)^{1/2}(T - T_{\mathbf{x}})(T + \lambda)^{-1/2}\}^{-1}\right\|_{\mathcal{L}(\mathcal{H}_K)}$$

$$\leq \left\|\sqrt{T}(T + \lambda)^{-1/2}\right\|_{\mathcal{L}(\mathcal{H}_K)} \sum_{i=1}^\infty \Omega(\lambda, \mathbf{z})^n$$

$$\text{(By spectral theorem)} \quad \leq \frac{1}{1 - \Omega(\lambda, \mathbf{z})} = 2. \tag{28}$$

On the other hand,

$$\Omega(\lambda, \mathbf{z})^2 = \langle(T + \lambda)^{-1}(T - T_{\mathbf{x}}), ((T + \lambda)^{-1}(T - T_{\mathbf{x}}))^*\rangle_{\mathcal{L}_2(\mathcal{H}_K)}$$

$$\leq \left\|(T + \lambda)^{-1}(T - T_{\mathbf{x}})\right\|_{\mathcal{L}_2(\mathcal{H}_K)}^2 = \Theta(\lambda, \mathbf{z})^2.$$

As a result, we have $\Omega(\lambda, \mathbf{z}) \leq \frac{1}{2}$ with probability greater than $1 - \delta_1$ as long as $n > N_1(\delta_1)$.

**Step 3.2**: probabilistic bound on $\left\|(T + \lambda)^{-1/2}(A_{\mathbf{z}}^*\psi(\mathbf{y}) - T_{\mathbf{x}}\mu_{\mathcal{H}_K})\right\|_{\mathcal{H}_K}$. Let $\xi_3 : \mathcal{Z} \to \mathcal{H}_K$ be the random variable

$$\xi_3(x, y) = (T + \lambda)^{-1/2}K_x(\psi(y) - \mu_{\mathcal{H}_K}(x)).$$

Via the same reasoning in the proof of Thm. 5 in [20], we have $\|\xi_3\|_{\mathcal{H}_K} \leq \sqrt{\frac{\kappa M}{\lambda}} = \frac{H_3}{2}$ and $\mathbb{E}[\|\xi_3\|_{\mathcal{H}_K}^2] \leq M\mathcal{N}(\lambda) = \sigma_3^2$. From our assumptions and Thm. 9 we know for each $\epsilon_3$ and $\delta_3$ there exists $N_3(\delta_3, \epsilon_3)$ such that

$$\left\|(T + \lambda)^{-1/2}(A_{\mathbf{z}}^*\psi(\mathbf{y}) - T_{\mathbf{x}}\mu_{\mathcal{H}_K})\right\|_{\mathcal{H}_K} \leq \epsilon_3 \tag{29}$$

with probability greater than $1 - \delta_3$ as long as $n > N_3(\delta_3, \epsilon_3)$.

Linking bounds (23), (25), (26), (28), and (29) we obtain that for every $\epsilon_1, \epsilon_2, \epsilon_3 > 0$ and $\delta_1, \delta_2, \delta_3 > 0$ there exists $N = \max\{N_1(\delta_1), N_2(\delta_2, \epsilon_2), N_3(\delta_3, \epsilon_3)\}$ such that for each $n > N$,

$$\mathcal{E}_s[\mu_{\mathbf{z}}^\lambda] - \mathcal{E}_s[\mu_{\mathcal{H}_K}] \leq 3[\mathcal{A}(\lambda) + \frac{\epsilon_2^2}{\lambda} + 4\epsilon_3^2]$$

with probability greater than $1 - \delta_1 - \delta_2 - \delta_3$. This means that for any $\epsilon > 0$ and fixed $\lambda$

$$\lim_{n \to 0} p\left(\mathcal{E}_s[\mu_{\mathbf{z}}^\lambda] - \mathcal{E}_s[\mu_{\mathcal{H}_K}] > 3\mathcal{A}(\lambda) + \epsilon\right) = 0 \tag{30}$$

From [25] we know

$$\lim_{\lambda \to 0} \mathcal{A}(\lambda) = 0. \tag{31}$$

Combining (30) and (31) we can conclude that as long as $\lambda$ decreases to 0, $\mathcal{E}_s[\mu_{\mathbf{z}}^\lambda]$ converges to $\mathcal{E}_s[\mu_{\mathcal{H}_K}]$ in probability. $\qquad\square$

**Theorem 5.** *Assume Hypothesis 1 and Hypothesis 2 in [20] and our Assumption 1 hold. With the conditions in Thm. 3, we assert that if $\lambda_n$ decreases to 0 sufficiently slowly,*

$$\mathcal{E}_s[\widehat{\mu}_{\lambda_n,n}] - \mathcal{E}_s[\mu'] \to 0 \tag{32}$$

*in probability as $n \to \infty$.*

*Proof.* The proof follows directly from Thm. 2, Thm. 3, Thm. 7, Corollary 2 and Thm. 10. □