[Reviews · NeurIPS 2016]

Reviewer 1

Summary

The paper proposes a vector-valued regression formulation of the kernel Bayes rule (KBR), and propses two imporovements of the KBR as consequences; one is the thresholding approach in regularized operator inversion, and the other is posterior regularization. The propsed methods are applied to nonlinear filtering problems, and better results are reported.

Qualitative Assessment

The proposed regularization formulation of kernel Bayes rule is novel as far as I know, while it is based on the existing regression formulation of the conditional kernel mean. In my opinion, the strongest advantage of this paper is the posterior regularization, which has potential to improve the exising KBR methods significantly, and this is a good contribution to the field. The followig are my specific comments. - The posterior regularization in Eq.(10) is a general form, and I think there are many ways of introducing (x_i,t_i). In the two examples in Section 5, specific ways are described. But it would be more useful, if the authors discuss in Section 4 various possible methods for defining the regularization terms or (x_i,t_i) in different situations. - It is not clear how the pKBR (theresholding) preforms differently from the original KBR. More discussions or theoretical analysis are needed. - Theorem 3 assumes the finite space, as in the previous literature. This is practically a very strong assumption, and further theoretical deveopment will be needed, while I agree that this is not an easy task. - The following comments are minor: -- line 136: is this arg min_\mu E[\mu]? -- In proposition 3, y_{m+1},...,y_n must be t_{m+1},...,t_n to match the symbols in Eq.(10).

Confidence in this Review

3-Expert (read the paper in detail, know the area, quite certain of my opinion)


Reviewer 2

Summary

The paper provides a new framework to understand kernel Bayesian inference by showing that the posterior embedding can be viewed as the minimizer of a vector valued regression problem. This new formulation motivates the authors to propose a new regularization technique that is shown to run faster and has comparable accuracy as the regularization method proposed in the original kernel Bayes' rule paper.

Qualitative Assessment

This paper provides an interesting connection between kernel Bayesian inference and vector valued regression. Based on this, a new regularization method is provided to compute an approximation of the kernel embedding of the posterior distribution. Simulation results look promising, suggesting that the new method gains improvement over many existing methods. However, as a non expert, from reading the current introduction, I'm still confused about the motivation of using kernel Bayesian inference---in order to approximate the kernel embedding of the posterior, a sample of iid draws (x_i, y_i) from the joint distribution of the parameter/hidden variable (X in the paper) and data (Y in the paper) are assumed to be available. First, it is a highly non-trivial problem of obtaining samples (x_i)'s from the posterior. Second, given that we have already obtain those samples (x_i)'s from the posterior, we can already do Bayesian inference without using the kernel embedding, for example, approximate the posterior distribution by the empirical distribution, or compute the expectation of any function with respect to the posterior. To summarize this point, I think that It would be very helpful if the authors can explicitly state the advantage or usefulness of using kernel Bayesian inference---at least, not like the current motivation in the beginning of the introduction, which is because of the popularity of kernel methods and kernel embeddings. Some minor comments: 1. line 64, please explicitly define \phi first, although I understand that it should refer to the feature map. Besides, I think that the definition of the kernel embedding by the feature map only works when \phi(x) = k_X(x,\cdot), where k_X is the reproducing kernel (by looking at some previous literature such as [4], [6]). If this is the case, then it would be good to replace \phi(x) with k_X(x,\cdot), since otherwise some expressions such as < f, \phi(X) > in line 70 does not make any sense. 2. The current paper does not say anything about the choice of the kernel (only require characteristic kernels). How does the choice of the kernel affects the approximation errors of the embedding? In the simulations, what kernel are you using and how they are chosen?

Confidence in this Review

1-Less confident (might not have understood significant parts)


Reviewer 3

Summary

This paper proposes theoretical results characterizing posterior mean embeddings as the solution of a vector-valued regression problem. As in RegBayes which formulates a posterior distribution (via Bayes' rule) as the solution to an optimization problem, this paper proposes kRegBayes which does the same to kernel Bayes' rule of Fukumizu et al., 2013. For computing a conditional mean embedding operator, a new regularization technique is also proposed and is shown to be faster and consistent.

Qualitative Assessment

# Justifications of the ratings Considering the complexity of this work which builds upon a number of existing technical papers, the authors did an excellent job in the writing. The writing is very clear and precise. The paper is self-contained. A potential academic impact from this work is from the optimizational formulation of kernel Bayesian inference which increases understanding and allows one to also add a regularization to the inferred posterior embedding. This can open up a new research direction for the kernel Bayes' rule as well as other techniques related to conditional mean embedding. # Questions 1. Section 3.1 mainly deals with how to estimate the loss in (3) without using an i.i.d. joint sample to approximate the expectation (as we do not have it). What is mysterious to me is the absorption of \mu into f. Originally in (3), \mu is what we seek. Now in (4), the loss in (3) is rewritten in such a way that the original \mu (a conditional mean embedding operator) is now in f, and a new \mu_{(X,Y)} appears (a joint embedding). I guess in the end f is treated as if it does not have \mu. Could you please elaborate a bit on what is going on here? 2. Line 152: f is assumed to be in a product RKHS. Is it really in the space? f looks like an unbounded function. What further conditions (e.g., on H_X and H_Y, or the domains) do we need for this to be true? 3. In theorem 3, does |\mathcal{X} \times \mathcal{Y}| < \infty mean the domains \mathcal{X}, \mathcal{Y} are discrete and not just that \mathcal{X} and \mathcal{Y} are bounded? If \mathcal{X}, \mathcal{Y} are indeed discrete, isn't this result fairly restrictive? Related: What are \mathcal{X}, \mathcal{Y} in the experiments? 4. Section 4 discusses a possibility of adding a regularization to the posterior embedding. I am not sure why the regularization (second term) in (10) is a sensible one. The description states that this is to encourage the posterior to be concentrated at some specified points. Why and when do we want this? I understand that this is just one way to do regularization under this framework. If there are more, what are others? # Minor comments * The proofs in the appendix are quite technical. It would be much better if the paper contains a short proof sketch for each new result. * Reference [14] has a wrong order of coauthors? * In section 2.2 and others, if I understand correctly \mu_{(YX)} is the same as C^{\pi}_{YX}. It might make it a bit more readable to just use only C^{\pi}_{YX}. * Line 136: arg sup should be arg inf. * Line 234: By (30), you mean (10)? ------------ after the rebuttal ---------- I read the authors' response.

Confidence in this Review

2-Confident (read it all; understood it all reasonably well)


Reviewer 4

Summary

The authors propose an alternative formulation of the Kernel Bayes' Rule as a vector-valued optimisation problem. They use thresholding regularisation and show consistency of the relevant estimators for finite spaces. They show how additional regularisation of the posterior embedding can be incorporated into their formulation and demonstrate that the resulting algorithm is more efficient than existing approaches on selected examples.

Qualitative Assessment

This paper ties together several lines of work, offering an interesting perspective on the Kernel Bayes' Rule. It is a valuable contribution, both illuminating the Kernel Bayes' Rule and having potential practical impact. The contribution is clearly explained and placed into the context of existing work. The theoretical results presented in the paper are sound, but I am concerned about their applicability. I understand that the crucial step is Theorem 3, which only applies to finite input spaces. I expect this not to be the case in most of the potential applications, in particular this is not the case in the examples used in Section 5. There are some other assumptions used without justification in the paper, such as that a certain function is in the relevant RKHS. Although those types of assumptions appear fairly standard in the literature on conditional embeddings, it would be useful to see some discussion of when they hold and if the resulting formulae can be expected to provide reasonable approximations when those assumptions are violated. Minor issues: [6] refers to the NIPS version, but results from the journal version are cited line 136 - argsup should be argmin equation (20) - f is already used in the statement of the theorem

Confidence in this Review

1-Less confident (might not have understood significant parts)


Reviewer 5

Summary

The paper deals with Bayesian inference by embedding the posterior density in a suitable RKHS Hilbert space. In this way, regularization is applied to the posterior, allowing for solving Bayesian problems in a computationally attractive manner. The proposed framework is applied to a nonlinear filtering problem.

Qualitative Assessment

The paper is written in a rather technical and sometimes obscure way, making it hardly understandable to those that are not expert in posterior embedding. Indeed, the way the framework is presented does not help the reader at all. Section 2, dedicated to the preliminaries, should help the reader in gathering the necessary background. However, this section starts with some hasty definitions and lacks of the proper notation setting. For instance, line 64 contains some notations that is never defined (precisely, \phi(X)). Now, I may understand that this notation is quite standard in this type of work, but, for instance, I could hardly get its meaning, as I am not an expert in the field. As a consequence, it becomes hard to understand even the basic concept of kernel embedding, which lies at the fundamentals of the whole paper. A Similar example is the definition of C_{(YX)Y} at line 101, which appears very abstract. Consequently, it is hard for me to give a critical comment on the paper. What I can only suggest is to try to make the paper more accessible to the reader. Perhaps, a paper of this type is more suited to a journal rather than a conference with page limits.

Confidence in this Review

1-Less confident (might not have understood significant parts)